# The basolateral amygdala complex and perirhinal cortex represent focal and peripheral states of information processing in rats

Francesca S Wong, Simon Killcross, R Fred Westbrook, Nathan M Holmes*

School of Psychology, University of New South Wales, Sydney, Australia

## eLife Assessment

This **important** Research Advance builds on the authors' previous work delineating the roles of the rodent perirhinal cortex and the basolateral amygdala in first- and second-order learning. The **convincing** results show that serial exposure of non-motivationally relevant stimuli influences how those stimuli are encoded within the perirhinal cortex and basolateral amygdala when paired with a shock. This manuscript will be interesting for researchers in cognitive and behavioral neuroscience.

**Abstract** We previously identified the roles of the basolateral amygdala complex (BLA) and perirhinal cortex (PRh) in sensory preconditioning in male and female rats (Wong et al., 2025). Here, we used variations of a sensory preconditioning protocol to test a general theory that the BLA and PRh represent focal and peripheral states of attention, respectively. We specifically tested predictions derived from the theory regarding when learning about a stimulus that signals danger will be disrupted by BLA or PRh infusions of the *N*-methyl-D-aspartate receptor (NMDAR) antagonist, DAP5. Consistent with the theory, the effects of these infusions depended on the novelty/familiarity of the conditioned stimulus, as well as the manner in which it was paired with foot shock. When a stimulus was novel, its conditioning required NMDAR-activation in the BLA and not the PRh, regardless of whether the stimulus and shock were presented contiguously or separated in time. When a pre-exposed and, thereby, familiar stimulus was presented contiguously with shock, its conditioning again required NMDAR-activation in the BLA and not the PRh. However, when a pre-exposed stimulus was indirectly paired with shock – because it was associatively activated at the time of shock or separated from the shock by another stimulus – its conditioning required NMDAR-activation in the PRh and not the BLA. These findings are discussed in relation to theories of information processing that distinguish between focal and peripheral states of attention/memory, and past studies that have examined the substrates of learning and memory in the PRh and BLA.

## Introduction

Sensory preconditioning is a laboratory protocol that can be used to examine how the mammalian brain processes sensory and emotional information. It typically involves three stages. In stage 1, rats are exposed to pairings of two relatively harmless stimuli, S2 and S1 (e.g. a sound and light). In stage 2, rats are exposed to pairings of one of these stimuli, the S1 (e.g. the light), with foot shock. Finally, in stage 3, rats display fear or defensive responses (e.g. freezing) when tested with the S1 that had been paired with shock and the S2 that had *never* been paired with shock. Importantly, the fear of S2 is not due to generalization from S1 or any innately aversive properties of the S2, as it is absent

*For correspondence: nathan.holmes@unsw.edu.au

Sent for Review 25 June 2025
Preprint posted 28 June 2025
Reviewed preprint posted 26 August 2025
Reviewed preprint revised 09 December 2025
Version of Record published 18 February 2026

among controls exposed to explicitly unpaired presentations of the relevant stimuli in either stage 1 or stage 2 (**Holmes et al., 2013**; **Holmes and Westbrook, 2017**; **Kikas et al., 2021**; **Michalscheck et al., 2021**; **Parkes and Westbrook, 2010**; **Rizley and Rescorla, 1972**; **Wong et al., 2019**; **Wong et al., 2025**). Instead, where it is observed, fear of the S2 reflects integration of the sensory and emotional information encoded in stages 1 and 2. That is, the S2-S1 (sensory) association formed in stage 1 is integrated with the S1-shock (emotional) association formed in stage 2 to drive fear responses during presentations of the S2 alone in stage 3.

Over the past 10 years, our laboratory has examined the neural substrates of the different types of information that rats encode in this sensory preconditioning protocol. Our work has focused on two regions of the medial temporal lobe – the perirhinal cortex (PRh) and basolateral amygdala complex (BLA) – and revealed that the involvement of these two regions can be quite distinct. For example, the S2-S1 association that forms in stage 1 is encoded through activation of $N$-methyl-D-aspartate receptors (NMDARs) in the PRh, not the BLA, whereas the S1-shock association that forms in stage 2 is encoded through activation of NMDARs in the BLA, not the PRh (**Bauer et al., 2002**; **Campeau et al., 1992**; **Fanselow and Kim, 1994**; **Miserendino et al., 1990**; **Holmes et al., 2013**; **Holmes et al., 2018**; **Rodrigues et al., 2001**; see also **Bang and Brown, 2009**; **Gewirtz and Davis, 1997**; **Phillips and LeDoux, 1995**; **Romanski and LeDoux, 1992a**; **Romanski and LeDoux, 1992b**; **Wilensky et al., 2006**). In a recent pair of studies, we extended these findings in two ways. First, we showed that S1 does not just form an association with shock in stage 2; it also mediates an association between S2 and the shock. Thus, S2 enters testing in stage 3 already conditioned, able to elicit fear responses (**Wong et al., 2019**). Second, we showed that this mediated S2-shock association requires NMDAR-activation in the PRh, as well as communication between the PRh and BLA (**Wong et al., 2025**). These findings raise two critical questions: (1) *why* is the PRh engaged for mediated conditioning of S2 but not for direct conditioning of S1; and (2) more generally, *what* determines whether the BLA and/or PRh is engaged for conditioning of the S1 and/or S2?

One explanation for the different substrates of conditioning to S2 and S1 in our sensory preconditioning protocol is that the two stimuli access different states of attention/memory which are differentially supported by the BLA and PRh. That is, during stage 2 of our protocol, the BLA may function like a focal state of attention in processing information about the presented S1 and foot shock. By contrast, because the PRh encodes the S2-S1 pre-exposures in stage 1, this region may function like a peripheral state of attention in processing information about the S2 that is retrieved from long-term memory. Such a characterization explains why the direct S1-shock association requires activation of NMDARs in the BLA but not the PRh (S1 and shock co-occur in the focus of attention); and the mediated S2-shock association requires activation of NMDARs in the PRh, as well as communication between the PRh and BLA (S2 conditions in the periphery of attention as the shock occurs in its focus). However, it also implies that the substrates of conditioning to S2 will depend on how the S2-S1 and S1-shock pairings are presented. For example, different types of arrangements may influence the substrates of conditioning to S2 by influencing its novelty and/or its predictive value at the time of the shock, on the supposition that familiar stimuli are processed in the periphery of attention and, thereby, the PRh (**Bogacz and Brown, 2003**; **Brown and Banks, 2015**; **Brown and Bashir, 2002**; **Martin et al., 2013**; **McLelland et al., 2014**; **Morillas et al., 2017**; **Murray and Wise, 2012**; **Robinson et al., 2010**; **Suzuki and Naya, 2014**; **Voss et al., 2009**; **Yang et al., 2023**), whereas novel stimuli are processed in the focus of attention and, thereby, the amygdala (**Holmes et al., 2018**; **Qureshi et al., 2023**; **Roozendaal et al., 2006**; **Rutishauser et al., 2006**; **Schomaker and Meeter, 2015**; **Wright et al., 2003**). This can be achieved by presenting the S2-S1 and S1-shock pairings together as S2-S1-shock sequences. Under such circumstances, conditioning to the S2 should cease to occur in the periphery of attention as it is novel and no longer requires retrieval of information from long-term memory. Instead, this conditioning should occur in the focus of attention and, thereby, require activation of NMDARs in the BLA and not the PRh (e.g. **Williams-Spooner et al., 2022**).

Different types of arrangements may also influence the substrates of conditioning to S2 as a function of its 'distance' or temporal separation from the shock. The supposition here is that shock/danger and its immediate antecedents are processed in the focus of attention (and, thereby, the amygdala), regardless of whether the antecedents are novel or familiar, whereas stimuli that are temporally separated from the shock *can be* processed in the periphery of attention (and, thereby, the PRh) and *are more likely to be* processed there if they are familiar. This can be assessed by pre-exposing rats to

**Table 1.** Designs and predictions for each experiment in the series.

| Experiment | Training | | Predictions | |
|---|---|---|---|---|
| | Stage 1 | Stage 2 | For S2 | For S1 |
| 1A | S2-S1 | 🔵S1-shock | Disrupted | Intact |
| 1B | S2-S1 | 🔴S1-shock | Intact | Disrupted |
| 2A | … | 🔵S2-S1-shock | Intact | Intact |
| 2B | … | 🔴S2-S1-shock | Disrupted | Disrupted |
| 3A | S2-S1 | 🔵S2-S1-shock | Disrupted | Intact |
| 3B | S2-S1 | 🔴S2-S1-shock | Intact | Disrupted |
| 4A | S2-S1 | 🔵S2-[trace]-shock | Intact | ▨ |
| 4B | S2-S1 | 🔴S2-[trace]-shock | Disrupted | ▨ |

Notes. 🔵=PRh infusion of DAP5; 🔴=BLA infusion of DAP5. In Experiments 1A, 1B, 3A, and 3B, S2 is familiar and indirectly paired with shock, whereas S1 is familiar and directly paired with shock. In Experiments 2A and 2B, S2 is novel and indirectly paired with shock, whereas S1 is novel and directly paired with shock. In Experiments 4A and 4B, S2 is familiar and more directly paired with shock (compared to Experiments 3A and 3B).

S2-S1 pairings prior to conditioning with S2-S1-shock sequences. Under these circumstances, conditioning to the familiar S1 that is contiguous with shock should occur in the focus of attention and, thereby, require activation of NMDARs in the BLA and not the PRh, whereas conditioning to the familiar S2 that is separated from the shock (by S1) should occur in the periphery of attention and, thereby, require activation of NMDARs in the PRh, and not the BLA.

The present series of experiments used these variations of a sensory preconditioning protocol to test the implications of our theory that the BLA and PRh represent focal and peripheral states of attention, respectively. The first implication is that, in our standard sensory preconditioning protocol (S2-S1 pairings in stage 1, S1-shock pairings in stage 2), the substrates of conditioning to S2 and S1 will be doubly dissociable at the level of NMDARs in the PRh and BLA. That is, during the session of S1-shock pairings in stage 2, mediated conditioning of the absent S2 should occur in the periphery of attention and hence, require NMDAR-activation in the PRh but *not* the BLA, whereas direct conditioning of the presented S1 should occur in the focus of attention and hence, require NMDAR-activation in the BLA but not the PRh. This was supported by the initial experiments which found that a stage 2 infusion of the NMDAR antagonist, DAP5, into the PRh disrupts mediated conditioning of S2 but spares direct conditioning of S1 (Experiment 1A), whereas a stage 2 infusion of DAP5 into the BLA disrupts direct conditioning of S1 but spares mediated conditioning of S2 (Experiment 1B). Subsequent experiments then used variations of this protocol to examine whether the engagement of NMDAR in the PRh or BLA for Pavlovian fear conditioning is influenced by the novelty/predictive value of the stimuli at the time of the shock (second implication of theory), as well as their distance or separation from the shock (third implication of theory; *Table 1*). Following the arguments above, we predicted that:

1. When S2 or S1 is novel and paired with shock, its conditioning will be disrupted by a DAP5 infusion into the BLA but unaffected by a DAP5 infusion into the PRh (Experiments 2A and 2B).
2. When S2 or S1 is familiar and directly paired (i.e. co-occurs) with shock, its conditioning will again be disrupted by a DAP5 infusion into the BLA but not the PRh (Experiments 1A, 1B, 3A, and 3B).
3. However, when S2 or S1 is familiar and indirectly paired with shock – because it is associatively activated at the time of shock or separated from the shock by another stimulus – its conditioning will be unaffected by a DAP5 infusion into the BLA but disrupted by a DAP5 infusion into the PRh (Experiments 1A, 1B, 3A, and 3B).

## Results

### Experiments 1A and 1B – Encoding of the mediated S2-shock association requires NMDAR-activation in the PRh but not the BLA; Encoding of the direct S1-shock association requires NMDAR-activation in the BLA but not the PRh

We propose that the BLA and PRh represent focal and peripheral states of attention/memory, respectively. The first implication of this proposal is that, during stage 2 of our standard sensory preconditioning protocol, the substrates of conditioning to the absent S2 and presented S1 will be doubly dissociable at the level of NMDARs in the PRh and BLA: mediated conditioning of the absent S2 should occur in the periphery of attention and hence, require NMDAR-activation in the PRh but *not* the BLA, whereas direct conditioning of the presented S1 should occur in the focus of attention and hence, require NMDAR-activation in the BLA but not the PRh. Experiments 1A and 1B tested this implication. The design of these experiments is shown in *Figure 1A*. The two experiments differed only in that we assessed the involvement of the PRh in Experiment 1A and the BLA in Experiment 1B. In each experiment, two groups of rats were exposed to a session of S2-S1 pairings in stage 1 and, 24 hr later, a session of S1-shock pairings in stage 2. Immediately prior to the conditioning session in stage 2, rats in each experiment received an infusion of DAP5 or vehicle only into either the PRh (Experiment 1A) or BLA (Experiment 1B). Finally, all rats were tested with presentations of the S2 alone and S1 alone in stage 3. Based on our past findings and the reasoning given above, we expected that mediated conditioning of the S2 would require activation of NMDARs in the PRh (*Wong et al., 2025*) but not the BLA, whereas the direct conditioning of the S1 would require activation of NMDARs in the BLA but not the PRh (e.g. *Rodrigues et al., 2001*; *Williams-Spooner et al., 2022*). Hence, we anticipated that the stage 2 infusion of DAP5 into the PRh would disrupt the test level of freezing to S2 without affecting the test level of freezing to S1, and that the stage 2 infusion of DAP5 into the BLA would disrupt the test level of freezing to S1 without affecting the test level of freezing to S2.

### Experiment 1A – NMDAR-activation in the PRh is required for acquisition of the mediated S2-shock association but not the direct S1-shock association

#### Conditioning

The baseline levels of freezing during conditioning and test sessions were low (<10%) and did not differ between groups (largest $F_{(1,19)} = 2.876$; p=0.106). Conditioning of the S1 was successful. Freezing to S1 increased across the four S1-shock pairings in stage 2 ($F_{(1,19)} = 74.708$; p<0.001; $n_p^2$=0.797; 95% CI: [1.658, 2.717]). The rate of this increase did not differ between the two groups ($F_{(1,19)} = 0.008$; p=0.930), and there was no significant between-group differences in overall freezing to the S1 ($F_{(1,19)} = 1.803$; p=0.195).

#### Test

*Figure 1B* shows the mean (± SEM) levels of freezing to the S2 alone (left panel) and S1 alone (right panel) during testing in Experiment 1A, averaged across the eight presentations of each stimulus. Both groups showed equivalent levels of freezing to the directly conditioned S1 ($F_{(1,19)} = 1.423$; p=0.248), indicating that the PRh infusion of DAP5 had no effect on this conditioning. However, rats that received a PRh infusion of DAP5 (Group DAP5) in stage 2 exhibited significantly less freezing to the preconditioned S2 than rats that received a PRh infusion of vehicle only (Group VEH; $F_{(1,19)} = 7.136$; p=0.015; $n_p^2$=0.273; 95% CI: [0.176, 1.447]).

### Experiment 1B – NMDAR-activation in the BLA is required for acquisition of the direct S1-shock association but not the mediated S2-shock association

#### Conditioning

The baseline levels of freezing during conditioning and test sessions were low (<10%) and did not differ between groups (largest $F_{(1,26)} = 1.590$; p=0.219). Conditioning of the S1 was successful. Freezing to S1 increased across the four S1-shock pairings in stage 2 ($F_{(1,26)} = 104.494$; p<0.001; $n_p^2$=0.801; 95% CI: [1.633, 2.456]). The rate of this increase did not differ between groups ($F_{(1,26)} = 0.259$; p=0.615); however, there was a significant between-group difference in overall freezing to

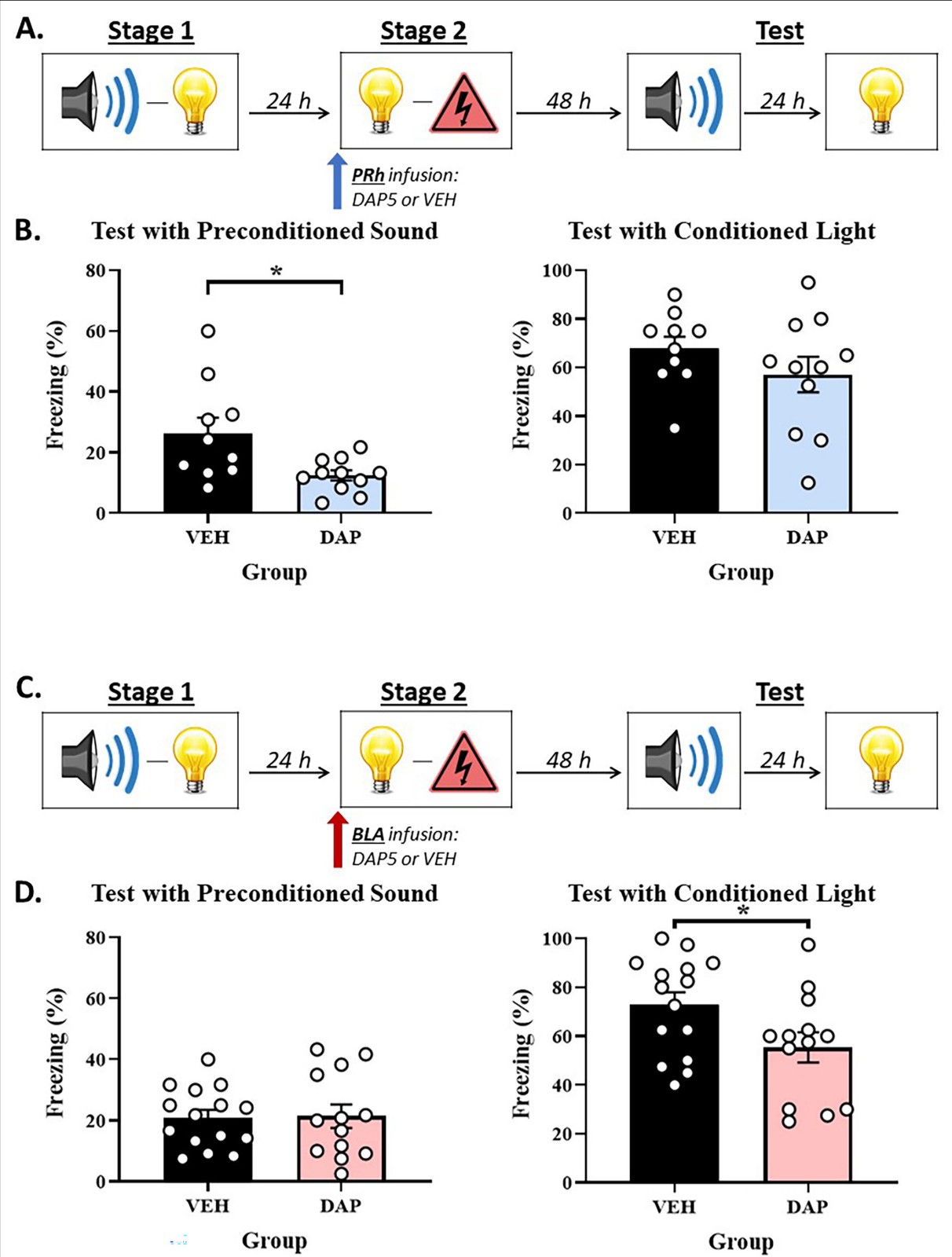

**Figure 1.** The neural substrates of the mediated S2-shock and direct S1-shock associations that form in stage 2 of sensory preconditioning are doubly dissociable at the level of *N*-methyl-D-aspartate receptor (NMDAR)-activation in the perirhinal cortex (PRh) and basolateral amygdala complex (BLA). (**A, C**) Schematics of the protocols used to assess the involvement of NMDAR in acquisition of freezing to S2 and S1 in the PRh (Experiment 1A) and BLA (Experiment 1B). The protocols differed only in whether rats received an infusion of the NMDAR antagonist, DAP5, into the PRh (in A) or BLA (in

*Figure 1 continued on next page*

*Figure 1 continued*

C). (**B**) Test results showing that a stage 2 infusion of DAP5 into the PRh disrupts freezing to S2 without affecting freezing to S1. (**D**) Test results showing that a stage 2 infusion of DAP5 into the BLA disrupts freezing to S1 without affecting freezing to S2. The bars in each histogram show the mean level of freezing in each group across repeated presentations of S2 or S1 under conditions of extinction. The error bars show the standard error of the mean, and the overlaid data points represent the mean level of freezing for individual rats. The final group sizes were n=10 for Group VEH and n=11 for Group DAP5 in Experiment 1A, and n=15 for Group VEH and n=13 for Group DAP5 in Experiment 1B.

the S1 ($F_{(1,26)}$ = 6.068; p=0.021; $n_p^2$=0.189; 95% CI: [0.094, 1.046]), indicating that the BLA infusion of DAP5 disrupted conditioning to the S1.

## Test

*Figure 1D* shows the mean (± SEM) levels of freezing to the S2 alone (left panel) and S1 alone (right panel) during testing in Experiment 1B, averaged across the eight presentations of each stimulus. Both groups showed equivalent levels of freezing to the preconditioned S2 ($F_{(1,26)}$ = 0.013; p=0.910). However, rats that received a BLA infusion of DAP5 (Group DAP5) in stage 2 exhibited significantly less freezing to the directly conditioned S1 than rats that received a BLA infusion of vehicle only (Group VEH; $F_{(1,26)}$ = 4.775; p=0.038; $n_p^2$=0.155; 95% CI: [0.040, 1.308]).

Experiment 1A replicated previous findings that a stage 2 infusion of DAP5 into the PRh disrupts the acquisition of sensory preconditioned freezing to S2 without affecting the acquisition of freezing to S1 (*Wong et al., 2025*). By contrast, Experiment 1B showed that a BLA infusion of DAP5 disrupts the acquisition of freezing to S1 (replication) without affecting the acquisition of sensory preconditioned freezing to S2 (new result). Taken together, these findings indicate that the involvement of NMDARs in direct and mediated conditioning is doubly dissociable at the levels of the BLA and PRh. The direct conditioning of a stimulus that co-terminates with shock requires activation of NMDARs in the BLA but does not require activation of NMDARs in the PRh. Conversely, the mediated conditioning of a stimulus representation that is retrieved from long-term memory (and, hence, indirectly paired with shock) requires activation of NMDARs in the PRh but occurs independently of NMDAR-activation in the BLA. These findings are exactly consistent with the predictions derived from our proposal that the BLA and PRh represent focal and peripheral states of attention/memory, respectively. The BLA represents a focal state and, thereby, encodes the directly conditioned S1-shock association, whereas the PRh represents a peripheral state and, thereby, encodes the indirect or mediated S2-shock association.

## Experiments 2A and 2B – The roles of the PRh and BLA in Pavlovian fear conditioning when the S2-S1 and S1-shock pairings are combined into an S2-S1-shock sequence

A second implication of our proposal is that the engagement of NMDAR in the PRh for conditioning to S2 will be influenced by its novelty/familiarity at the time of the shock. We addressed this implication by combining the S2-S1 and S1-shock pairings into S2-S1-shock sequences that were administered in a single training session (Experiments 2A and 2B). The consequence of doing so is that, in contrast to the previous experiments in which the S2 and S1 were both familiar at the time of their conditioning with foot shock, here the S2 and S1 were both novel at the time of their conditioning with foot shock. Immediately prior to the session containing the S2-S1-shock sequences, rats received an infusion of DAP5 or vehicle only into either the PRh (Experiment 2A) or BLA (Experiment 2B). Finally, all rats were tested with presentations of the S2 alone and S1 alone in stage 3. We hypothesized that, because S2 was novel and presented in sequence with S1 and shock in stage 2, it would be processed in a focal state of attention rather than the periphery. Hence, whereas conditioning to a representation of the pre-exposed S2 in the previous experiment required NMDAR-activation in the PRh but not the BLA, we predicted that conditioning to the novel S2 in this experiment would require NMDAR-activation in the BLA but not the PRh. Specifically, we expected that the stage 2 infusion of DAP5 into the PRh would have no effect on the test level of freezing to either S2 or S1, and that the stage 2 infusion of DAP5 into the BLA would disrupt the test level of freezing to both S2 and S1.

## Experiment 2A – When the S2-S1 and S1-shock pairings are combined into S2-S1-shock sequences, the PRh is no longer engaged for conditioning of the S2

### Conditioning

The baseline levels of freezing during conditioning and test sessions were low (<10%) and did not differ between groups (largest $F_{(1,15)}$ = 3.026; p=0.102). Conditioning was successful. Freezing to the S2 and S1 increased linearly across the four S2-S1-shock sequences in stage 2 (smaller $F_{(1,15)}$ = 47.193; p<0.001; $n_p^2$=0.759; 95% CI: [1.453, 2.760]). The linear × group interaction was significant for freezing to the S2 ($F_{(1,15)}$ = 5.117; p<0.039; $n_p^2$=0.254; 95% CI: [0.056, 1.868]), but not for freezing to the S1 ($F_{(1,15)}$ = 0.009; p=0.926). This suggests that the PRh infusion of DAP5 slowed acquisition of freezing to the S2 without affecting freezing to the S1. However, there were no significant between-group differences in overall freezing to the S2 ($F_{(1,15)}$ = 4.125; p = 0.06) or S1 ($F_{(1,15)}$ = 2.726; p = 0.120).

### Test

*Figure 2B* shows the mean (± SEM) levels of freezing to the S2 alone (left panel) and S1 alone (right panel) during drug-free testing in Experiment 2A, averaged across the eight presentations of each stimulus. Both groups showed equivalent levels of freezing to the S2 ($F_{(1,15)}$ = 1.292; p=0.274) and S1 ($F_{(1,15)}$ = 0.838; p=0.374).

## Experiment 2B – When the S2-S1 and S1-shock pairings are combined into S2-S1-shock sequences, the BLA is engaged for conditioning of the S2

### Conditioning

The baseline levels of freezing during conditioning and test sessions were low (<10%) and did not differ between groups (largest $F_{(1,15)}$ = 0.576; p=0.460). Conditioning was again successful. Freezing to the S2 and S1 (smaller $F_{(1,15)}$ = 62.211; p<0.001; $n_p^2$=0.806; 95% CI: [1.093, 1.902]) increased linearly across the four S2-S1-shock sequences in stage 2. The linear × group interaction was significant for freezing to the S2 ($F_{(1,15)}$ = 9.477; p=0.008; $n_p^2$=0.387; 95% CI: [0.337, 1.852]) and S1 ($F_{(1,15)}$ = 12.771; p=0.003; $n_p^2$=0.460; 95% CI: [0.548, 2.167]), suggesting that the BLA infusion of DAP5 slowed acquisition of freezing to both stimuli. However, there were no significant between-group differences in overall freezing to the S2 ($F_{(1,15)}$ = 2.999; p=0.104) or S1 ($F_{(1,15)}$ = 3.747; p = 0.072).

### Test

*Figure 2D* shows the mean (± SEM) levels of freezing to the S2 alone (left panel) and S1 alone (right panel) during drug-free testing in Experiment 2B, averaged across the eight presentations of each stimulus. Rats that received a BLA infusion of DAP5 (Group DAP5) in stage 2 exhibited significantly less freezing to both the S2 ($F_{(1,15)}$ = 8.660; p=0.010; $n_p^2$=0.366; 95% CI: [0.349, 2.182]) and S1 ($F_{(1,15)}$ = 12.089; p=0.003; $n_p^2$=0.446; 95% CI: [0.507, 2.114]) compared to the vehicle-infused controls.

Experiments 2A and 2B showed that combining the S2-S1 and S1-shock pairings into S2-S1-shock sequences altered the roles of the PRh and BLA in conditioning to the S2. In the previous experiments, the S2-S1 (stage 1) and S1-shock (stage 2) pairings were separated by 24 hr, and acquisition of freezing to S2 required NMDAR-activation in the PRh but not the BLA. Here, the acquisition of freezing to S2 (and S1) across S2-S1-shock sequences required NMDAR-activation in the BLA but not the PRh. These findings are consistent with our proposal that the BLA and PRh function like different states of attention/memory. The BLA represents a focal state and, thus, processes information about novel stimuli that are presented in a sequence that co-terminates with shock; hence, it was engaged for conditioning to both the S2 and S1 in the current experiment. By contrast, the PRh represents a peripheral state and maintains the trace or representation of a pre-exposed stimulus; hence, it was not engaged for conditioning to either the S2 or S1 in the current experiment.

## Experiments 3A and 3B – The roles of the PRh and BLA in Pavlovian fear conditioning when rats are pre-exposed to S2-S1 pairings prior to conditioning with S2-S1-shock sequences

A third implication of our proposal is that pre-exposure can alter the way that stimuli are processed in the focus/periphery of attention and, thereby, the substrates of their conditioning in the PRh and BLA.

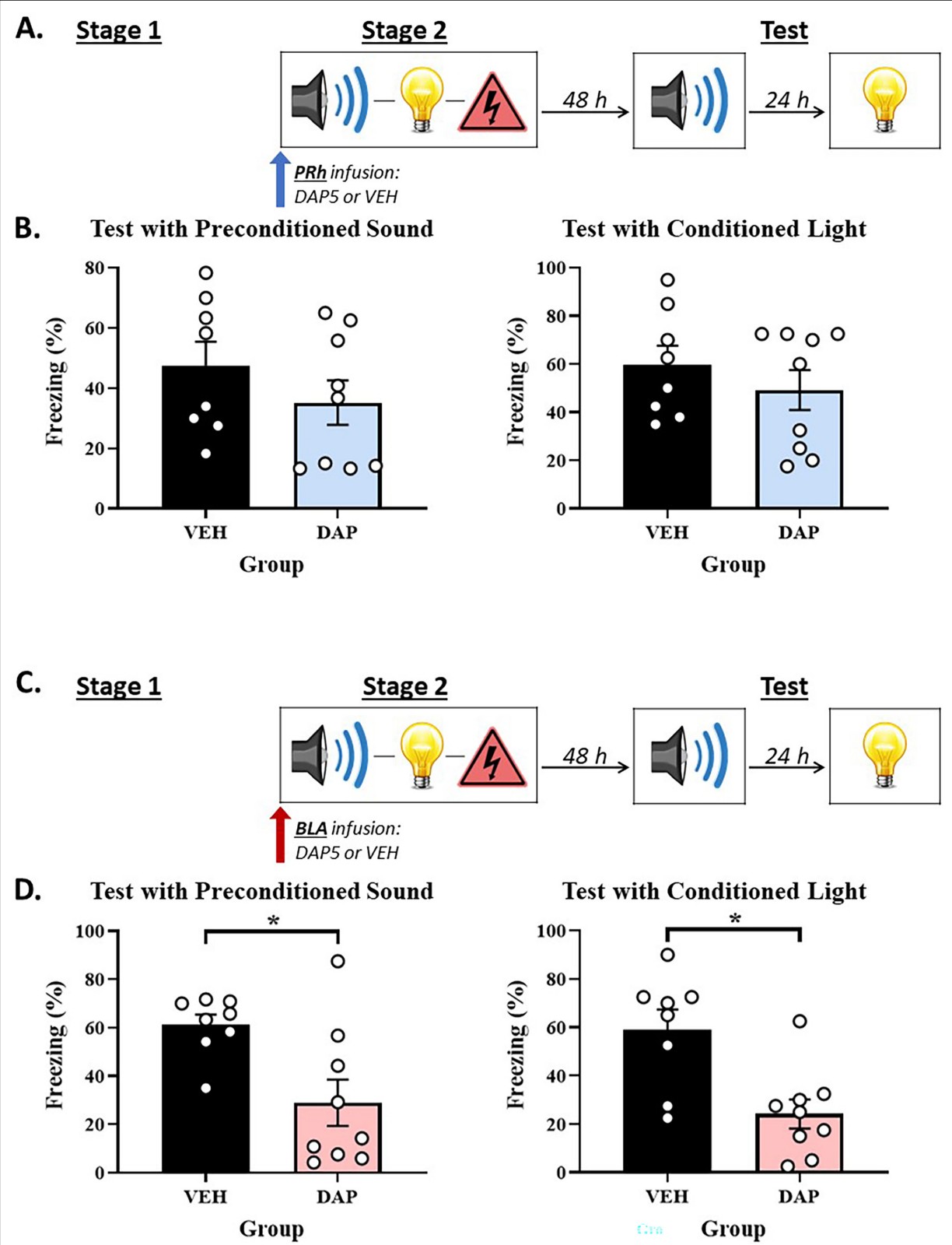

**Figure 2.** Combining the S2-S1 and S1-shock pairings into S2-S1-shock sequences alters the substrates of conditioning to the S2 in the perirhinal cortex (PRh) and basolateral amygdala complex (BLA). (**A, C**) Schematics of the protocols used to assess when *N*-methyl-D-aspartate receptor (NMDAR) is involved in acquisition of freezing to S2 and S1 in the PRh (Experiment 2A) and BLA (Experiment 2B). The protocols differed only in whether rats received an infusion of the NMDAR antagonist, DAP5, into the PRh (in A) or BLA (in C). (**B**) Test results showing that a stage 2 infusion of DAP5 into the PRh has

*Figure 2 continued on next page*

*Figure 2 continued*

no effect on levels of freezing to either S2 or S1. (**D**) Test results showing that a stage 2 infusion of DAP5 into the BLA disrupts levels of freezing to both the S2 and S1. The final group sizes were n=8 for Group VEH and n=9 for Group DAP5 in Experiment 2A, and n=8 for Group VEH and n=9 for Group DAP5 in Experiment 2B.

The additional implication is based on decades of work showing that pre-exposure to a stimulus can reduce its rate of conditioning (*Holmes and Harris, 2009*; *Holmes and Harris, 2010*; *Kennedy et al., 2021*; *Leung et al., 2013a*; *Leung et al., 2013b*; *Lubow, 1973*; *Lubow and Moore, 1959*) while increasing the context dependence of any responding that develops (*Holmes and Westbrook, 2013*; *Rosenberg et al., 2011*; *Westbrook et al., 2000*). We addressed this implication by pre-exposing rats to S2-S1 pairings in stage 1 followed by conditioning with S2-S1-shock sequences in stage 2. We specifically examined whether pre-exposing rats to S2-S1 pairings reduces processing of the S2 in the focus of attention when it is presented as part of S2-S1-shock sequences in stage 2; i.e., whether it reduces the ability of S2 to access the focus of attention (unlikely given the S1 results in Experiments 1A and 1B) and/or accelerates the rate at which S2 transitions to the periphery when it is replaced by presentation of the S1. If so, the substrates of conditioning to S2 in the current experiments should differ from the substrates of conditioning to S2 in Experiments 2A and 2B. That is, whereas conditioning of the novel S2 in Experiments 2A and 2B required NMDAR-activation in the BLA and not the PRh, conditioning of the pre-exposed S2 in the current experiments should require NMDAR-activation in the PRh and not the BLA.

The design of Experiments 3A and 3B is shown in *Figure 3*. Briefly, rats were exposed to a session of S2-S1 pairings in stage 1 and, 24 hr later, a session of S2-S1-shock sequences in stage 2. Immediately prior to the latter session, rats received an infusion of DAP5 or vehicle only into either the PRh (Experiment 3A) or BLA (Experiment 3B). Finally, all rats were tested with presentations of the S2 alone and S1 alone in stage 3. We hypothesized that pre-exposure to S2-S1 pairings prior to conditioning with S2-S1-shock sequences would *not* affect the substrates of directly conditioned fear to S1; i.e., just as the direct conditioning of a pre-exposed S1 in Experiments 1A and 1B required NMDAR-activation in the BLA but not the PRh, here again we expected that direct conditioning of the pre-exposed S1 would require NMDAR-activation in the BLA but not the PRh. Such a result would suggest that, with the level of pre-exposure used here, the S1 accesses the focus of attention; hence, its conditioning requires activation of NMDAR in the BLA and not the PRh. By contrast, we hypothesized that pre-exposure to S2-S1 pairings prior to conditioning with S2-S1-shock sequences would alter the substrates of conditioning to S2; i.e., whereas conditioning to the novel S2 in the previous experiments required NMDAR-activation in the BLA but not the PRh; here, we expected that conditioning of the pre-exposed S2 would require NMDAR-activation in the PRh but not the BLA. Such a result would suggest that a pre-exposed S2 more rapidly transitions from the focus of attention to its periphery when it is replaced by S1; hence, its conditioning ceases to require activation of NMDAR in the BLA and now requires activation of NMDAR in the PRh.

## Experiment 3A – When rats are pre-exposed to S2-S1 pairings prior to conditioning with S2-S1-shock sequences, the PRh is engaged for acquisition of freezing to S2 but not S1

### Conditioning

The baseline levels of freezing during conditioning and test sessions were low (<10%) and did not differ between groups (largest $F_{(1,22)}$ = 3.143; p=0.090). Conditioning was successful. Freezing to the S2 and S1 increased across the four S2-S1-shock sequences in stage 2 (smaller $F_{(1,22)}$ = 110.127; p<0.001; $n_p^2$=0.833; 95% CI: [1.877, 2.802]). The rate of this increase did not differ between groups (largest $F_{(1,22)}$ = 1.397; p=0.250), and there were no significant between-group differences in overall freezing to the S2 and S1 (largest $F_{(1,22)}$ = 0.620; p=0.439).

### Test

*Figure 3B* shows the mean (± SEM) levels of freezing to the S2 alone (left panel) and S1 alone (right panel) during testing in Experiment 3A, averaged across the eight presentations of each stimulus. Both groups showed equivalent levels of freezing to the directly conditioned S1 ($F_{(1,22)}$ = 0.226; p=0.639).

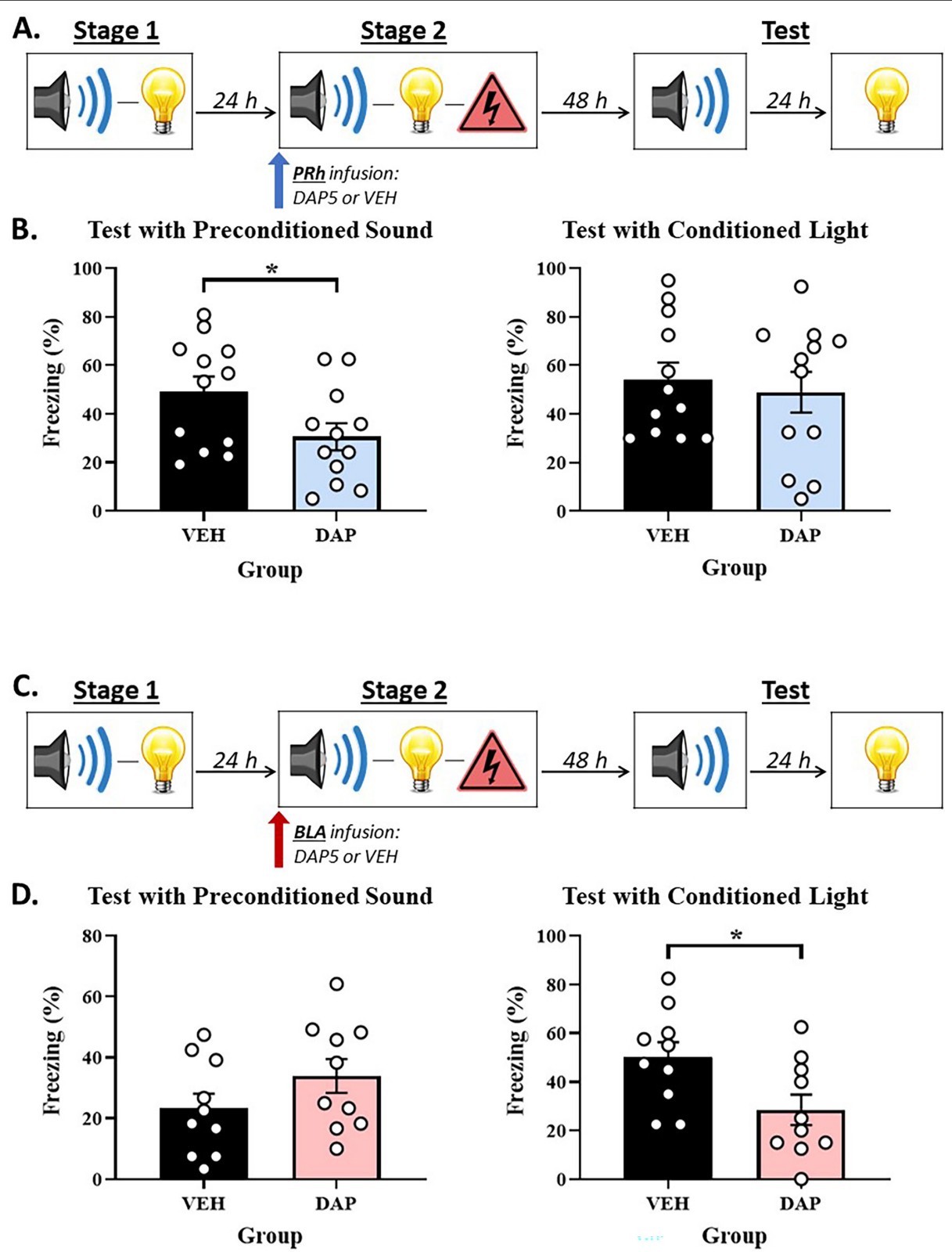

**Figure 3.** Pre-exposing rats to S2-S1 pairings prior to a session of S2-S1-shock sequences re-engages *N*-methyl-D-aspartate receptor (NMDAR) in the perirhinal cortex (PRh) for conditioning of the S2. (**A, C**) Schematics of the protocols used to assess when NMDARs are involved in acquisition of freezing to S2 and S1 in the PRh (Experiment 3A) and basolateral amygdala complex (BLA) (Experiment 3B). The protocols differed only in whether rats received an infusion of the NMDAR antagonist, DAP5, into the PRh (in A) or BLA (in C). (**B**) Test results showing that a stage 2 infusion of DAP5 into the

*Figure 3 continued on next page*

*Figure 3 continued*

PRh disrupts freezing to S2 without affecting freezing to S1. (**D**) Test results showing that a stage 2 infusion of DAP5 into the BLA disrupts freezing to S1 without affecting freezing to S2. The final group sizes were n=12 for Group VEH and n=12 for Group DAP5 in Experiment 3A, and n=10 for Group VEH and n=10 for Group DAP5 in Experiment 3B.

However, rats that received a PRh infusion of DAP5 (Group DAP5) in stage 2 exhibited significantly less freezing to the S2 than rats that received a PRh infusion of vehicle only (Group VEH; $F_{(1,22)}$ = 4.644; p=0.042; $n_p^2$=0.174; 95% CI: [0.025, 1.330]).

## Experiment 3B – When rats are pre-exposed to S2-S1 pairings prior to conditioning with S2-S1-shock sequences, the BLA is engaged for acquisition of freezing to S1 but not S2

### Conditioning

The baseline levels of freezing during conditioning and test sessions were low (<10%) and did not differ between groups (largest $F_{(1,18)}$ = 2.045; p=0.170). Conditioning was again successful. Freezing to the S2 and S1 increased across the four S2-S1-shock sequences in stage 2 (smaller $F_{(1,18)}$ = 36.409; p<0.001; $n_p^2$=0.669; 95% CI: [1.035, 2.141]). The rate of this increase did not differ between groups (largest $F_{(1,18)}$ = 1.474; p=0.240), and there were no significant between-group differences in overall freezing to the S2 ($F_{(1,18)}$ = 1.056; p = 0.318) and S1 ($F_{(1,18)}$ = 0.071; p = 0.793).

### Test

*Figure 3D* shows the mean (± SEM) levels of freezing to the S2 alone (left panel) and S1 alone (right panel) during testing in Experiment 3B, averaged across the eight presentations of each stimulus. Both groups showed equivalent levels of freezing to the S2 ($F_{(1,18)}$ = 2.078; p=0.167). However, rats that received a BLA infusion of DAP5 (Group DAP5) in stage 2 exhibited significantly less freezing to the directly conditioned S1 than rats that received a BLA infusion of vehicle (Group VEH; $F_{(1,18)}$ = 5.899; p=0.026; $n_p^2$=0.247; 95% CI: [0.114, 1.571]).

Experiments 3A and 3B showed that pre-exposure can alter the way that stimuli are processed in the focus/periphery of attention and, thereby, the substrates of their conditioning in the PRh and BLA. In the preceding experiments (2A and 2B) where rats were conditioned with S2-S1-shock sequences in the absence of any stimulus pre-exposure, the acquisition of freezing to S1 and S2 required NMDAR-activation in the BLA but not the PRh. By contrast, in the present experiments where rats were conditioned with S2-S1-shock sequences 24 hr after having been pre-exposed to S2-S1 pairings, the substrates of conditioning to S1 and S2 were distinct: acquisition of freezing to S1 required NMDAR-activation in the BLA but not the PRh, whereas acquisition of freezing to S2 required NMDAR-activation in the PRh but not the BLA. These findings are consistent with our proposal that the BLA and PRh function like different states of attention. The BLA functions like a focal state and, thus, processes information about stimuli that are directly paired with shock; hence, it was engaged for conditioning to the pre-exposed S1 which co-terminated with foot shock but was not engaged for conditioning to the pre-exposed S2 which was separated from the shock by S1. By contrast, the PRh represents a peripheral state and, as such, maintains a trace/representation of a pre-exposed stimulus when it is replaced by another stimulus; hence, it was not engaged for conditioning to the pre-exposed S1 which co-terminated with foot shock but was engaged for conditioning to the pre-exposed S2 which was separated from the shock by S1.

## Experiments 4A and 4B – The roles of the PRh and BLA in Pavlovian fear conditioning when rats are pre-exposed to S2-S1 pairings prior to conditioning with S2-[trace]-shock sequences

In Experiments 3A and 3B, conditioning of the pre-exposed S1 required NMDAR-activation in the BLA and not the PRh, whereas conditioning of the pre-exposed S2 required NMDAR-activation in the PRh and not the BLA. We attributed these findings to the fact that the pre-exposed S2 was separated from the shock by S1 during conditioning of the S2-S1-shock sequences in stage 2; hence, at the time of the shock, S2 was no longer processed in the focal state of attention supported by the BLA; instead, it was processed in the peripheral state of attention supported by the PRh.

Experiments 4A and 4B employed a modification of the protocol used in Experiments 3A and 3B to examine whether a pre-exposed S1 influences the processing of a pre-exposed S2 across conditioning with S2-S1-shock sequences. The design of these experiments is shown in *Figure 4A*. Briefly, in each experiment, two groups of rats were exposed to a session of S2-S1 pairings in stage 1 and, 24 hr later, a session of S2-[trace]-shock pairings in stage 2, where the duration of the trace interval was equivalent to that of S1 in the preceding experiments. Immediately prior to the trace conditioning session in stage 2, one group in each experiment received an infusion of DAP5 or vehicle only into either the PRh (Experiment 4A) or BLA (Experiment 4B). Finally, all rats were tested with presentations of the S2 alone in stage 3. If the substrates of conditioning to S2 are determined only by the amount of time between presentations of this stimulus and foot shock in stage 2, the results obtained in Experiments 4A and 4B should be the same as those obtained in Experiments 3A and 3B: acquisition of freezing to S2 will require activation of NMDARs in the PRh and not the BLA. If, however, the presence of S1 in the preceding experiments (Experiments 3A and 3B) accelerated the rate at which processing of S2 transitioned from the focus of attention to its periphery, the results obtained in Experiments 4A and 4B will differ from those obtained in Experiments 3A and 3B. That is, in contrast to the preceding experiments where acquisition of freezing to S2 required NMDAR-activation in the PRh and not the BLA, here acquisition of freezing to S2 should require NMDAR-activation in the BLA but not the PRh.

## Experiment 4A – When rats are exposed to S2-S1 pairings in stage 1 and trace conditioning of S2 in stage 2, the PRh is NOT required for encoding of the S2-shock association

### Conditioning
The baseline levels of freezing during conditioning and test sessions were low (<10%) and did not differ between groups (largest $F_{(1,27)}$ = 1.095; p=0.305). Conditioning was successful. Freezing to the S2 increased across the four S2-[trace]-shock pairings in stage 2 ($F_{(1,27)}$ = 120.496; p<0.001; $n_p^2$=0.817; 95% CI: [1.966, 2.871]). The rate of this increase did not differ between the two groups ($F_{(1,27)}$ = 0.332; p=0.569), and there was no significant difference between them in overall freezing to the S2 ($F_{(1,27)}$ = 0.323; p=0.575).

### Test
*Figure 4B* shows the mean (± SEM) levels of freezing to the S2 alone (left panel) during testing in Experiment 4A, averaged across the eight presentations of this stimulus. Both groups showed equivalent levels of freezing to the trace conditioned S2 ($F_{(1,27)}$ = 1.505; p=0.230).

## Experiment 4B – When rats are exposed to S2-S1 pairings in stage 1 and trace conditioning of S2 in stage 2, the BLA is required for encoding of the S2-shock association

### Conditioning
The baseline levels of freezing during conditioning and test sessions were low (<10%) and did not differ between groups (largest $F_{(1,24)}$ = 3.823; p=0.062). Conditioning was successful. Freezing to the S2 increased across the four S2-[trace]-shock pairings in stage 2 ($F_{(1,24)}$ = 47.799; p<0.001; $n_p^2$=0.666; 95% CI: [1.069, 1.979]). The rate of this increase did not differ between the two groups ($F_{(1,24)}$ = 3.421; p=0.077); however, there was a significant between-group difference in overall freezing to the S2 ($F_{(1,24)}$ = 7.040; p=0.014; $n_p^2$=0.227; 95% CI: [0.168, 1.348]).

### Test
*Figure 4D* shows the mean (± SEM) levels of freezing to the S2 alone (left panel) during testing in Experiment 4B, averaged across the eight presentations of this stimulus. Rats that received a BLA infusion of DAP5 (Group DAP5) in stage 2 exhibited significantly less freezing to the trace conditioned S2 than rats that received a BLA infusion of vehicle only ($F_{(1,24)}$ = 7.390; p=0.012; $n_p^2$=0.235; 95% CI: [0.213, 1.554]).

Together with the results of the preceding experiments (3A and 3B), the present findings show that when a pre-exposed S2 is paired with shock, the substrates of its conditioning are determined by more than just the time between its offset and the shock: they are additionally determined by what

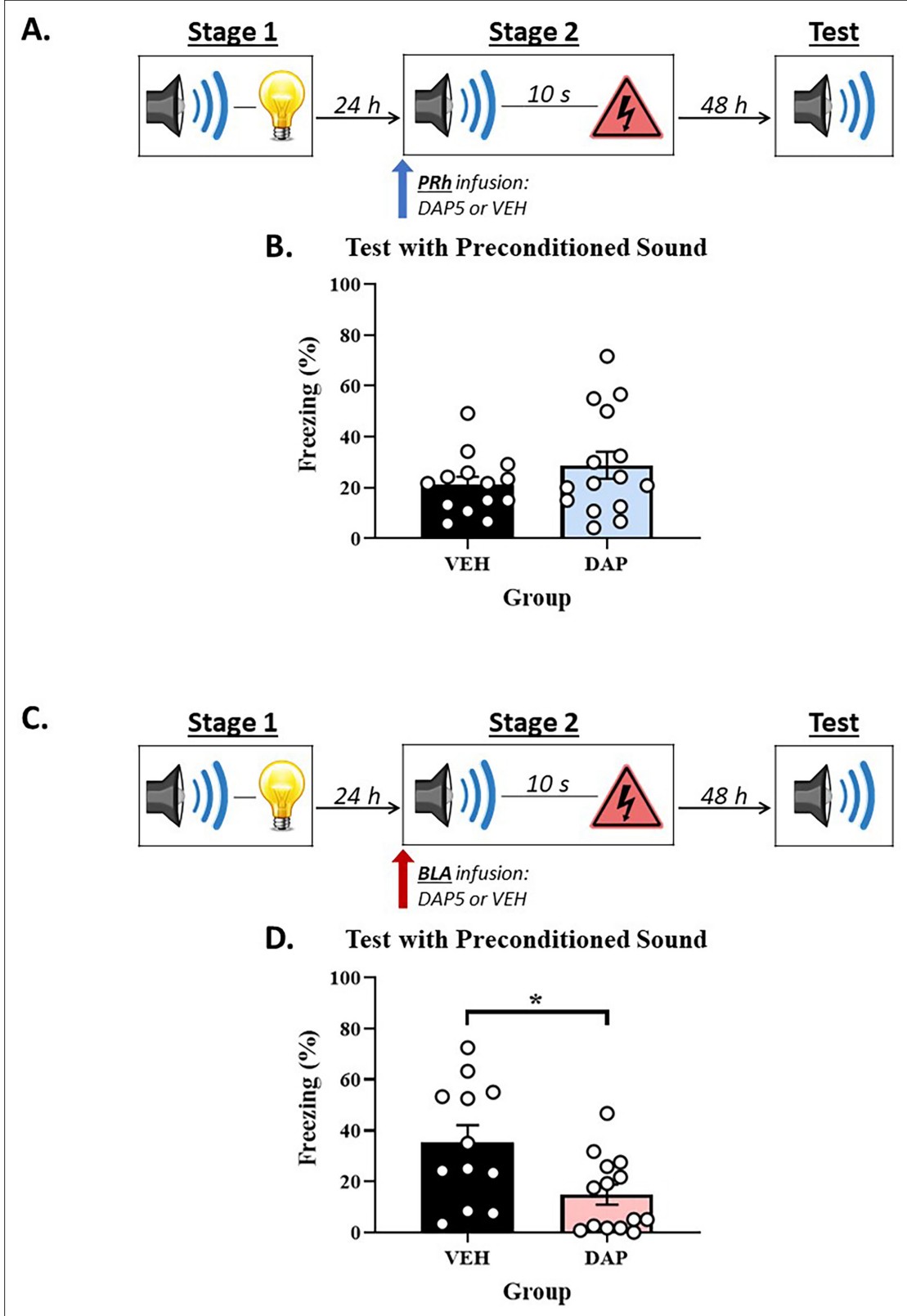

**Figure 4.** Pre-exposing rats to S2-S1 pairings prior to a session of S2-[trace]-shock pairings re-engages *N*-methyl-D-aspartate receptor (NMDAR) in the basolateral amygdala complex (BLA) for conditioning of the S2. (**A, C**) Schematics of the protocols used to assess when NMDARs are involved in acquisition of freezing to S2 and S1 in the perirhinal cortex (PRh) (Experiment 4A) and BLA (Experiment 4B). (**B, D**) Test results showing that a stage 2 infusion of DAP5 has no effect on the level of freezing to S2 when injected into the PRh (in B) but disrupts the level of freezing to S2 when injected into

*Figure 4 continued on next page*

*Figure 4 continued*

the BLA (in D). The final group sizes were n=14 for Group VEH and n=15 for Group DAP5 in Experiment 3A, and n=12 for Group VEH and n=14 for Group DAP5 in Experiment 3B.

happens in the interval between the two events. When the interval is blank, as it was in the present experiments (4A and 4B), the acquisition of freezing to S2 requires NMDAR-activation in the BLA, not the PRh. When the interval is occupied by a presentation of the pre-exposed S1, as it was in the preceding experiments (3A and 3B), the acquisition of freezing to S2 requires NMDAR-activation in the PRh, not the BLA. These findings can be understood in terms of our proposal that the PRh and BLA function like different states of attention: the presence of S1 in the preceding experiments (Experiments 3A and 3B) accelerated the rate at which processing of S2 transitioned to a peripheral state of attention; hence, its conditioning in those experiments required NMDAR-activation in the PRh and not the BLA. This and other implications of our proposal will be considered further in the Discussion.

## Discussion

This series of experiments used variations of a sensory preconditioning protocol in rats to test the implications of our proposal that the BLA and PRh represent different states of attention: the BLA functions like a focal state in processing stimuli that are physically present, regardless of their associative history, whereas the PRh functions like a peripheral state in supporting the traces or representations of pre-exposed stimuli that are not physically present. The first implication was that, in our standard sensory preconditioning protocol, the substrates of conditioning to S2 and S1 would be doubly dissociable at the level of NMDARs in the PRh and BLA: during the session of S1-shock pairings in stage 2, mediated conditioning of the absent S2 would occur in the periphery of attention and hence, require NMDAR-activation in the PRh but *not* the BLA, whereas direct conditioning of the presented S1 would occur in the focus of attention and hence, require NMDAR-activation in the BLA but not the PRh. In these experiments, rats were exposed to a session of S2-S1 pairings in stage 1 and, 24 hr later, to a session of S1-shock pairings in stage 2. We have previously shown that, across the session of S1-shock pairings, the S1 is directly associated with foot shock and mediates an association between the *absent-but-expected* S2 and foot shock. Here, we replicated our previous findings that formation of the direct S1-shock association requires activation of NMDARs in the BLA but not the PRh (*Williams-Spooner et al., 2022*; *Wong et al., 2025*), and extended these findings to show that formation of the mediated S2-shock association requires activation of NMDARs in the PRh but not the BLA. Specifically, we found that a BLA infusion of the NMDAR antagonist, DAP5, immediately prior to the training session in stage 2 disrupted freezing to the S1 at test without affecting the level of freezing to S2, and, conversely, a PRh infusion of DAP5 immediately prior to stage 2 disrupted freezing to the S2 at test without affecting the level of freezing to S1. These findings were taken to mean that the BLA regulates conditioning to a stimulus that is directly paired with danger and, hence, processed in the focus of attention, whereas the PRh regulates conditioning to a stimulus that is indirectly paired with danger and, hence, processed in a peripheral state of attention/memory. Consequently, blocking NMDAR in the BLA impaired direct conditioning of the S1 but spared mediated conditioning of the S2, and blocking NMDAR in the PRh impaired mediated conditioning of the S2 but spared direct conditioning of the S1.

The second implication of our proposal was that combining the S2-S1 and S1-shock pairings into S2-S1-shock sequences should selectively alter the substrates of conditioning to S2. Under these circumstances, the novel S2 would be more effectively maintained in the focus of attention along with the S1; hence, conditioning to both stimuli would occur in the BLA and not the PRh. This prediction was supported in Experiments 2A and 2B. Here, rats were exposed to S2-S1-shock sequences, and the acquisition of freezing to both S1 and S2 was disrupted by a DAP5 infusion into the BLA but unaffected by a DAP5 infusion into the PRh. That is, combining the S2-S1 pairings and S1-shock pairings into S2-S1-shock sequences had no impact on the substrates of conditioning to S1: as in Experiments 1A and 1B where the S2-S1 and S1-shock pairings were separated by 24 hr, acquisition of freezing to S1 required NMDAR-activation in the BLA but not the PRh. It did, however, alter the substrates of conditioning to S2: in contrast to Experiments 1A and 1B where acquisition of freezing to S2 required

NMDAR-activation in the PRh but not the BLA, here acquisition of freezing to S2 required NMDAR-activation in the BLA but not the PRh.

The remaining implications of our proposal concerned the effects of pre-exposing rats to S2-S1 pairings 24 hr prior to conditioning with S2-S1-shock sequences. We predicted that, relative to the results of Experiments 2A and 2B, pre-exposing rats to S2-S1 pairings would again alter the substrates of conditioning to S2 without affecting the substrates of conditioning to S1. Specifically, we predicted that at the end of each S2-S1-shock sequence in conditioning (i.e. the time of the shock), the pre-exposed (and thereby familiar) S2 would be processed in the periphery of attention while the S1 is processed in its focus: hence, conditioning to S2 would occur in the PRh and not the BLA, whereas conditioning to S1 would occur in the BLA and not the PRh. These predictions were supported in Experiments 3A and 3B. Rats were pre-exposed to S2-S1 pairings and, 24 hr later, exposed to S2-S1-shock sequences under the influence of a DAP5 infusion into the PRh or BLA. The test results revealed that the acquisition of freezing to S2 was disrupted by a DAP5 infusion into the PRh but unaffected by a DAP5 infusion into the BLA, whereas the acquisition of freezing to S1 was disrupted by a DAP5 infusion into the BLA but unaffected by a DAP5 infusion into the PRh. That is, pre-exposing rats to S2-S1 pairings prior to conditioning with S2-S1-shock sequences again had no effect on the substrates of conditioning to S1: as in Experiments 2A and 2B where rats received no pre-exposure, acquisition of freezing to S1 required NMDAR-activation in the BLA but not the PRh. It did, however, alter the substrates of conditioning to S2: in contrast to Experiments 2A and 2B where acquisition of freezing to the novel S2 required NMDAR-activation in the BLA but not the PRh, here acquisition of freezing to the pre-exposed S2 required NMDAR-activation in the PRh but not the BLA.

The results of Experiments 2A, 2B, 3A, and 3B provide support for our idea that the BLA and PRh function like focal and peripheral states of attention (respectively) by showing that the locus of conditioning in one region or the other is determined by novelty/familiarity of the stimulus (or whether it has been pre-exposed or not) and the nature of stimulus-shock pairings (whether they are direct or indirect). The results of Experiments 4A and 4B extended these findings by showing that the substrates of conditioning to a pre-exposed stimulus are also influenced by what happens in the period between presentations of the stimulus and shock. Here, rats were pre-exposed to S2-S1 pairings and, 24 hr later, exposed to S2-[trace]-shock pairings under the influence of a DAP5 infusion into the PRh or BLA. Thus, these experiments were like the preceding ones (Experiments 3A and 3B) in that S2 was separated from the shock by 10 s during its conditioning, but critically differed from the preceding experiments in the fact that the 10 s interval between the S2 and shock was *not* filled by a presentation of the S1. The consequences of leaving this interval blank were evident at test, where the pattern of results produced by infusions of DAP5 into the PRh and BLA was exactly opposite to that obtained in Experiments 3A and 3B. That is, whereas Experiments 3A and 3B showed that freezing to S2 was disrupted by a DAP5 infusion into the PRh but unaffected by a DAP5 infusion into the BLA, here the test level of freezing to S2 was disrupted by a DAP5 infusion into the BLA but unaffected by a DAP5 infusion into the PRh. These results were taken to mean that the substrates of conditioning to a pre-exposed S2 are influenced by more than just the interval between presentations of this stimulus and foot shock: it is also influenced by events that occur in that interval. If the interval is blank, the pre-exposed S2 may remain in a focal state of attention; hence, its conditioning requires NMDAR-activation in the BLA and not the PRh (Experiments 4A and 4B). If the interval is occupied by S1, the pre-exposed S2 more rapidly transitions to a peripheral state of attention; hence, its conditioning requires NMDAR-activation in the PRh and not the BLA (Experiments 3A and 3B).

Taken together, the results of the present study are generally consistent with theories of information processing that distinguish between focal and peripheral states of attention (e.g. *Atkinson and Shiffrin, 1968*; *Posner and Snyder, 1975*; *Shiffrin and Schneider, 1977*). Importantly, these theories were developed to explain the different characteristics of information processing; they were not developed to explain changes in the substrates of learning and memory with variations in the circumstances of training. They do, however, provide an excellent framework for thinking about the present findings. For example, according to one type of theory (*Wagner, 1981*), stimuli are represented as a series of elements that transition between states of activity and inactivity, and these transitions are governed by formal rules. Briefly, when a stimulus is presented, its elements are activated to the focal state of attention (A1) from which they decay to a peripheral state (A2) before returning to a state of inactivity. Importantly, the capacity of A1 is limited and much less than that of A2; stimuli processed

in A1 can prime their associates to A2; and the rate at which stimulus elements decay from A1 to A2 is influenced by their familiarity (a result of context-induced priming), as well as other stimuli that are present (i.e. the presence of other stimuli increases the processing demand on A1 and, thereby, the rate of information decay to A2). Given these features and our proposal that the BLA and PRh represent the focus and periphery of attention, a version of this theory can accommodate the full pattern of results in this study (*Holland, 1983*). When rats are exposed to S2-S1-shock sequences in the absence of any pre-exposure (Experiments 2A and 2B), the elements of both S2 and S1 overlap with those of the foot shock in the focal state of attention. Hence, conditioning to each stimulus requires activation of NMDAR in the BLA and not the PRh. When rats are conditioned with S1-shock pairings in stage 2 after pre-exposure to S2-S1 pairings in stage 1 (Experiments 1A and 1B), S1 elements are activated to the focus of attention and prime S2 elements to the peripheral state. Hence, conditioning to S1 requires NMDAR-activation in the BLA and not the PRh, whereas conditioning to S2 requires NMDAR-activation in the PRh and not the BLA. When rats are conditioned with S2-S1-shock sequences in stage 2 after pre-exposure to S2-S1 pairings in stage 1 (Experiments 3A and 3B), the S2 elements are initially processed in the focus of attention but are rapidly shunted to the periphery during presentations of the S1. Hence, conditioning to S1 requires NMDAR-activation in the BLA and not the PRh, whereas conditioning to S2 requires NMDAR-activation in the PRh and not the BLA. Finally, when rats are conditioned with S2-[trace]-shock pairings after pre-exposure to S2-S1 pairings in stage 1 (Experiments 4A and 4B), the S2 elements are initially processed in the focus of attention and naturally decay to the periphery during the trace interval. However, as the trace interval is relatively short, most S2 elements remain in the focal state at the time of foot shock; hence, conditioning to S2 requires NMDAR-activation in the BLA and not the PRh. With a longer trace interval, the theory predicts that there will be a greater accumulation of S2 elements in the peripheral state; hence, conditioning to S2 would cease to require NMDAR-activation in the BLA and now require NMDAR-activation in the PRh. This will be tested in future research.

An additional point to consider in relation to Experiments 3A, 3B, 4A, and 4B is the level of surprise that rats experienced following presentations of the familiar S2 in stage 2. Specifically, in Experiments 3A and 3B, S2 was followed by the expected S1 (low surprise), and its conditioning required activation of NMDA receptors in the PRh and not the BLA. By contrast, in Experiments 4A and 4B, S2 was followed by omission of the expected S1 (high surprise), and its conditioning required activation of NMDA receptors in the BLA and not the PRh. This raises the possibility that surprise, or prediction error, also influences the way that S2 is processed in focal and peripheral states of attention. When prediction error is low, S2 is processed in the peripheral state of attention; hence, learning under these circumstances requires NMDA receptor activation in the PRh and not the BLA. By contrast, when prediction error is high, S2 is preserved in the focal state of attention; hence, learning under these circumstances requires NMDA receptor activation in the BLA and not the PRh. The impact of prediction error on the processing of S2 could be assessed using two types of designs. In the first design, rats are pre-exposed to S2-S1 pairings in stage 1, and this is followed by S2-S3-shock pairings in stage 2. The important feature of this design is that, in stage 2, S2 is followed by surprise in omission of S1 and presentation of S3. Thus, if a large prediction error maintains processing of the familiar S2 in the BLA, we might expect that its conditioning in this design would require NMDA receptor activation in the BLA (in contrast to the results of Experiment 3B) and no longer require NMDA receptor activation in the PRh (in contrast to the results of Experiment 3A). In the second design, rats are pre-exposed to S2 alone in stage 1, and this is followed by S2-[trace]-shock pairings in stage 2. The important feature of this design is that, in stage 2, the S2 is not followed by the surprising omission of any stimulus. Thus, if a small prediction error shifts processing of the familiar S2 to the PRh, we might expect that its conditioning in this design would no longer require NMDA receptor activation in the BLA (in contrast to the results of Experiment 4B) but, instead, require NMDA receptor activation in the PRh (in contrast to the results of Experiment 4A). Future studies will use both designs to determine whether prediction error influences the processing of S2 in the focus versus periphery of attention and, thereby, whether learning about this stimulus requires NMDA receptor activation in the BLA or PRh.

Finally, the present findings add to a body of work which shows that the neural substrates of Pavlovian conditioned fear critically depend on the type of conditioning (*Keidar et al., 2023*; *Lay et al., 2018*; *Leake et al., 2024*; *Leidl et al., 2018*; *Williams-Spooner et al., 2019*, *Williams-Spooner et al., 2022*), and extend our previous work in sensory preconditioning by identifying how the

mediated S2-shock association is encoded at the level of the PRh and BLA. Specifically, our previous work showed that formation of this association requires neuronal activity in the PRh (*Wong et al., 2019*), as well as communication between this region and the BLA (*Wong et al., 2025*). Here, we have extended these findings by showing that NMDAR-activation is a critical component of the processing requirement in the PRh, but is not a component of the processing requirement in the BLA. Future work will identify the nature of the processing requirement in the BLA and exactly how the distinct signaling requirements in the PRh and BLA combine to generate the mediated S2-shock association.

In summary, the present study has shown that conditioning to a novel stimulus requires activation of NMDAR in the BLA but does not require activation of NMDAR in the PRh. By contrast, the substrates of conditioning to a pre-exposed stimulus are influenced by several factors, including its contiguity with shock and events interpolated between its presentations and shock. The overall pattern of results is consistent with our proposal that the BLA and PRh function like focal and peripheral states of attention/memory, respectively, and generally consistent with information processing theories which specify how events come to be represented in the focal and peripheral states (*Holland, 1983*; *Wagner, 1981*). Future research will test the additional implications of these theories and continue to examine commonalities and differences in the processes through which mediated and direct associations are encoded and stored in the medial temporal lobe.

## Materials and methods
### Subjects
Subjects were experimentally naïve male and female Long-Evans rats, obtained from a colony maintained by the Biological Resources Centre at the University of New South Wales. The rats were housed by sex in plastic cages (40 cm wide × 22 cm high × 67 cm long), with four rats per cage and food and water continuously available. The cages were located in an air-conditioned and humidity-controlled colony room maintained on a 12 hr light-dark cycle (lights on at 7 a.m. and off at 7 p.m.) at a temperature of approximately 21°C. All procedures were approved by the Animal Care and Ethics Committee at the University of New South Wales and conducted in accordance with the Australian code for the care and use of animals for scientific purposes (published by the National Health and Medical Research Council of Australia). Please note that these subject details (source, housing conditions, and animal maintenance) are exactly the same as those reported in past studies by our laboratory (e.g. *Kennedy et al., 2024*; *Wong et al., 2025*).

### Surgery
Prior to behavioral training and testing in each experiment, rats were surgically implanted with cannulas targeting either the PRh (Experiments 1A, 2A, 3A, and 4A) or BLA (Experiments 1B, 2B, 3B, and 4B). Rats were anesthetized using isoflurane, which was delivered in a steady stream of oxygen. The rat was then mounted onto a stereotaxic apparatus (David Kopf Instruments) and incisions made over the skull. Two holes were drilled through the skull and 26-gauge guide cannulas (Plastic Ones) were implanted into the brain, one in each hemisphere. For Experiments 1A, 2A, 3A, and 4A, the tips of the cannulas targeted the PRh at coordinates 4.30 mm posterior to Bregma, 5.00 mm lateral to the midline, 8.4 mm ventral to Bregma, and angled at approximately 9° (*Paxinos and Watson, 2007*). For Experiments 1B, 2B, 3B, and 4B, the tips of the cannulas targeted the BLA at coordinates 2.40 mm posterior to Bregma, 4.90 mm lateral to the midline, 8.40 mm ventral to Bregma (*Paxinos and Watson, 2007*). Guide cannulas were secured in place with four jeweler's screws and dental cement. A dummy cannula was kept in each guide cannula at all times except during drug infusions. Immediately after surgery, rats received a subcutaneous (SC) injection of a prophylactic dose (0.1 ml/kg) of Duplocillin (Merck & Co, NJ, USA). Rats were allowed 7 days to recover from surgery, during which they were monitored and weighed daily. Please note that these surgery details are exactly the same as those reported in our earlier studies (e.g. *Wong et al., 2019*; *Wong et al., 2025*).

### Drug infusions
In each experiment, the NMDA receptor antagonist, DAP5, or vehicle was infused bilaterally into the PRh or BLA. For these infusions, infusion cannulas were connected to 25 µl Hamilton syringes via polyethylene tubing. These syringes were fixed to an infusion pump (Harvard Apparatus). The infusion

procedure began by removing the dummy caps from the guide cannulas on each rat and inserting 33-gauge infusion cannulas in their place. The pump was programmed to infuse a total of 0.5 μl of DAP5 at a rate of 0.25 μl/min, which resulted in a total infusion time of 2 min. The infusion cannulas remained in place for an additional 2 min after the infusion was complete to allow for diffusion of the drug into the PRh or BLA tissue and, thereby, avoid reuptake when the infusion cannulas were withdrawn. This resulted in a total infusion time of 4 min. After the additional 2 min, the infusion cannulas were removed and replaced with dummy cannulas. The day prior to infusions, the dummy cannulas were removed and the infusion pump was activated to familiarize the rats with the procedure and thereby minimize any nonspecific effect of the procedure on the day of infusions. Please note that these drug and infusion details are exactly the same as those reported in the earlier study by *Wong et al., 2025*.

## Drugs

The NMDA receptor antagonist, DAP5 (Sigma, Australia), was prepared in the manner described by *Williams-Spooner et al., 2022*. Briefly, it was dissolved in ACSF to a concentration of 10 μg/μl and injected into the PRh or BLA as described above.

## Histology

After behavioral training and testing, rats were euthanized with a lethal dose of sodium pentobarbital. The brains were extracted and sectioned into 40 μm coronal slices. Every second slice was mounted onto a glass microscope slide and stained with cresyl violet. The placement of the cannula tip was determined under a microscope using the boundaries defined by *Paxinos and Watson, 2007*. Please note that the details for histology are exactly the same as those reported in the earlier studies by *Wong et al., 2019*; *Wong et al., 2025*.

Rats with misplaced cannulas were excluded from statistical analysis. *Figure 5* shows placement of the most ventral portion of these cannulas in the BLA for all rats that were included in the statistical analyses of data from Experiments 1B, 2B, 3B, and 4B. *Figure 6* shows placement of the most ventral portion of these cannulas in the PRh for all rats that were included in the statistical analyses of data from Experiments 1A, 2A, 3A, and 4A. The numbers of rats excluded from each experiment based on misplaced cannulas were 3 rats in Experiment 1A, 12 rats in Experiment 1B, 9 rats in Experiment 2A, 11 rats in Experiment 2B, 6 rats in Experiment 3A, 6 rats in Experiment 3B, 3 rats in Experiment 4A, and 5 rats in Experiment 4B. A further nine rats were excluded from all experiments due to issues with their head caps and/or health issues prior to the infusion day. The final *n*s for each group are shown in the figure legend for each experiment.

## Behavioral apparatus

All experiments were conducted in four identical chambers (30 cm wide × 26 cm long × 30 cm high). The side walls and ceiling were made of aluminum, and the front and back walls of clear plastic. The floor consisted of stainless steel rods, each 2 mm in diameter and spaced 13 mm apart (center-to-center). A waste tray containing bedding material was located under the floor. At the end of each session, the chambers were cleaned with water and any soiled bedding was removed from the waste trays and replaced with fresh bedding. Each chamber was enclosed in a sound- and light-attenuating wooden cabinet. The cabinet walls were painted black. A speaker and LED lights within a fluorescent tube were mounted onto the back wall of each cabinet. The speaker was used to deliver a 1000 Hz square-wave tone stimulus, presented at 75 dB when measured at the center of the chamber (digital sound meter: Dick Smith Electronics, Australia). The LED lights were used to deliver a flashing light stimulus, presented at 3.5 Hz. A custom-built, constant current generator was used to deliver a 0.5 s duration, 0.8 mA intensity shock to the floor of each chamber. Each chamber was illuminated with an infrared light source, and a camera mounted on the back wall of each cabinet was used to record the behavior of each rat. The cameras were connected to a monitor and a high-definition digital image recorder located in an adjacent room. This room also contained the computer that controlled stimulus and shock presentations through an interface and appropriate software (MATLAB, MathWorks, USA). Please note that the behavioral apparatus used for this study was exactly the same as that used in the earlier studies by *Wong et al., 2019*; *Wong et al., 2025* and a previous study by *Kennedy et al., 2024*.

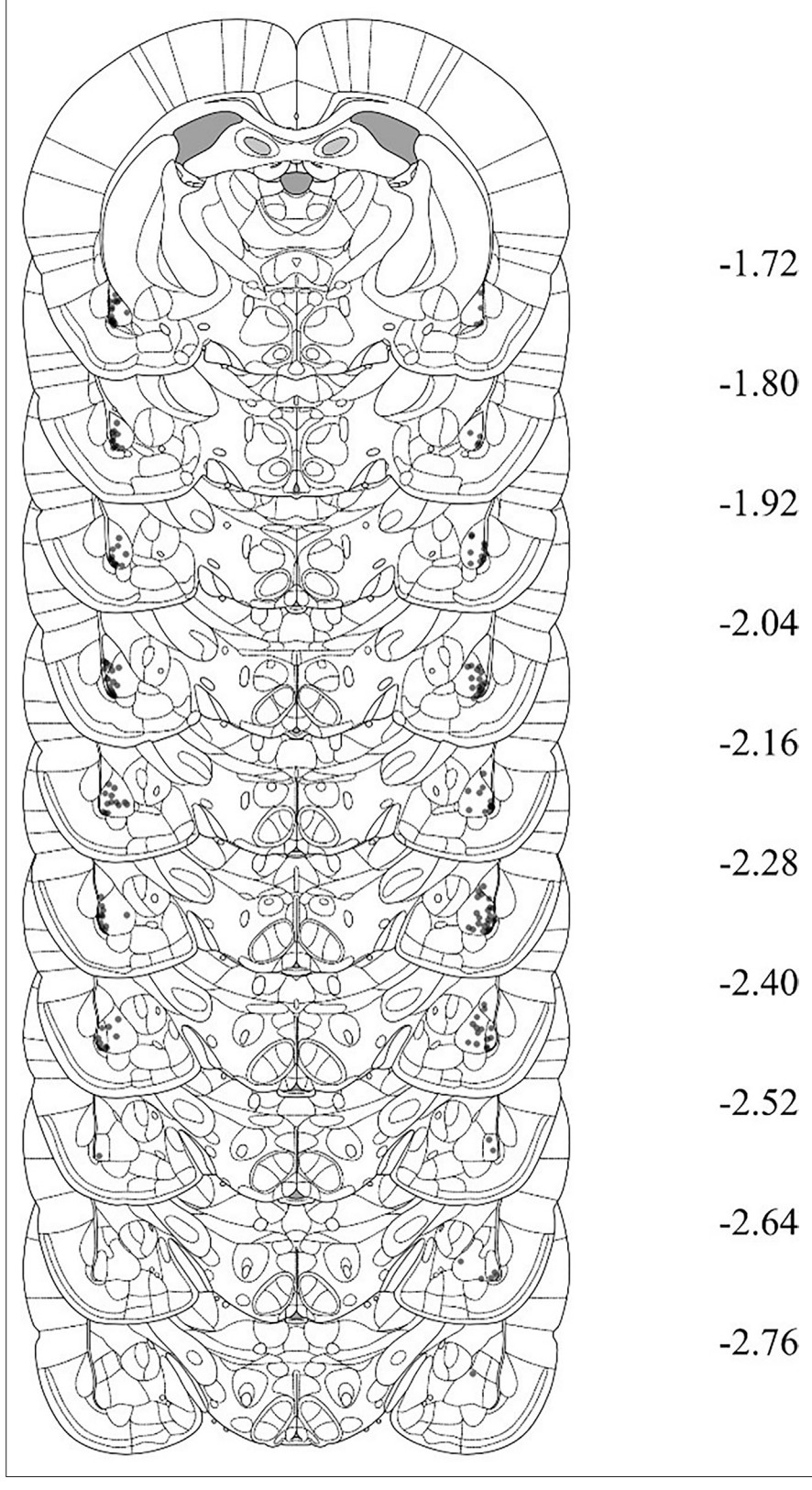

-1.72

-1.80

-1.92

-2.04

-2.16

-2.28

-2.40

-2.52

-2.64

-2.76

**Figure 5.** Cannula placements in the perirhinal cortex (PRh) for rats in Experiments 1B, 2B, 3B, and 4B. The most ventral portion of the cannulas is marked on coronal sections based on the atlas of *Paxinos and Watson, 2007*.

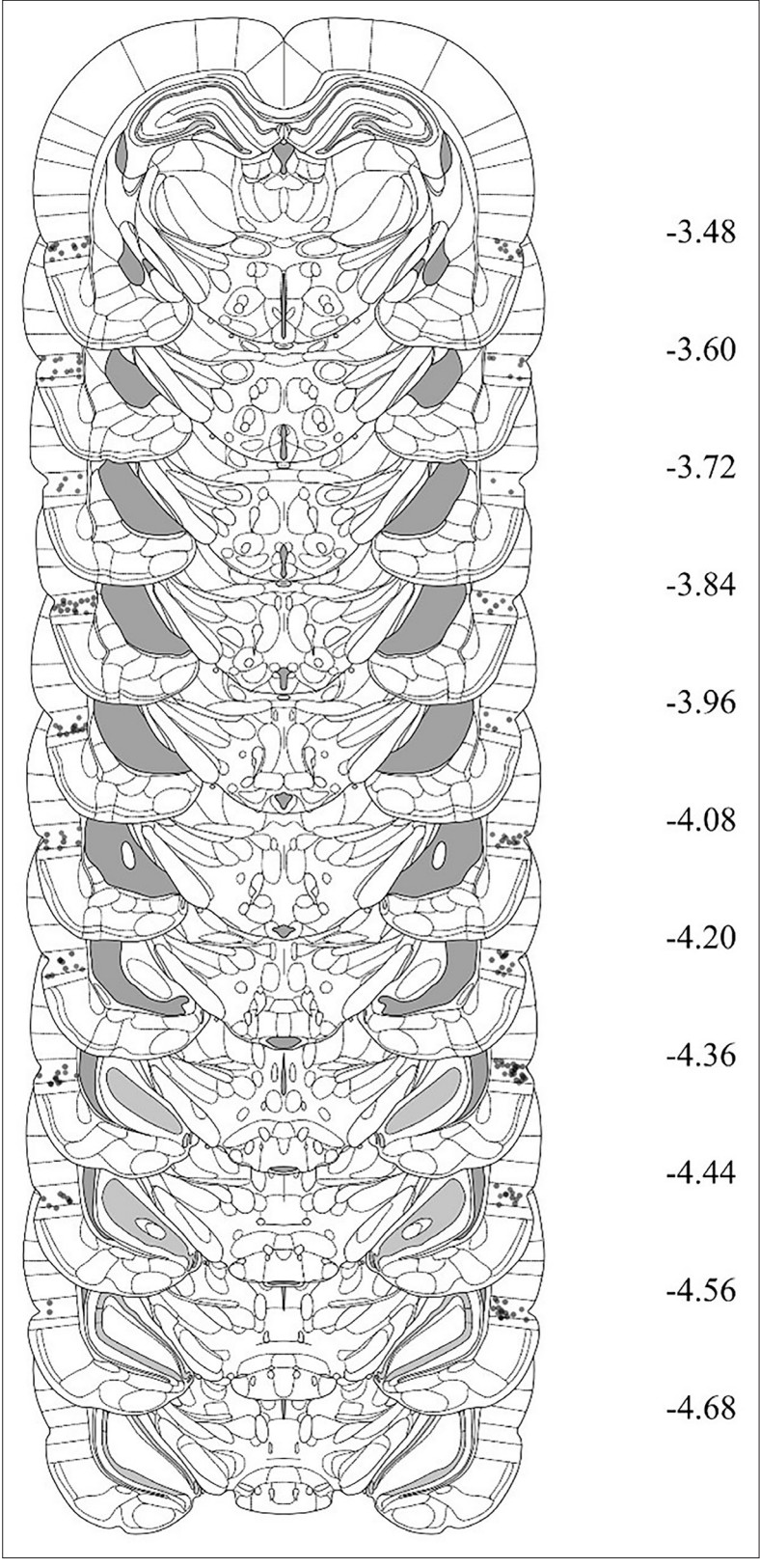

**Figure 6.** Cannula placements in the perirhinal cortex (PRh) for rats in Experiments 1A, 2A, 3A, and 4A. The most ventral portion of the cannulas is marked on coronal sections based on the atlas of *Paxinos and Watson, 2007*.

## Behavioral procedure

### Experiments 1A and 1B

#### Context exposure

On days 1 and 2, rats received two 20 min exposures to the chambers, one in the morning and the other approximately 3 hr later in the afternoon.

#### Stage 1: Sensory preconditioning

On day 3, rats were exposed to eight paired presentations of the stimuli designated S2 and S1. For half the rats, the S2 was a sound and the S1 was a light. For the remaining rats, the S2 was a light and the S1 was a sound. Each presentation of the S2 was 30 s in duration, each presentation of the S1 was 10 s in duration, and the stimuli were paired in such a way that the offset of S2 co-occurred with the onset of S1. The first S2 presentation occurred 5 min after rats were placed into the chambers, and the interval between each S2-S1 pairing was 5 min (measured from the offset of the S1 to onset of the next S2). After the last S1 presentation, rats remained in the chambers for an additional 1 min. They were then returned to their home cages.

#### Stage 2: First-order conditioning of the S1

On day 4, rats were exposed to four paired presentations of the S1 and foot shock. The first S1 presentation occurred 5 min after rats were placed in chambers, each 10 s S1 co-terminated with the 0.5 s foot shock, and the interval between the S1-shock pairings was 5 min. Rats remained in the chambers for an additional 1 min after the final S1-shock pairing and were then returned to their home cages.

#### Context extinction

On day 5, rats received two 20 min exposures to the chambers, one in the morning and the other 3 hr later in the afternoon. These exposures were intended to extinguish any freezing elicited by the context alone so that the level of freezing to the S2 and S1 could be assessed unconfounded by context-elicited freezing. On day 6, rats received a further 10 min extinction exposure to the context.

#### Testing

On day 6, approximately 2 hr after the context extinction session, rats were tested for their levels of freezing to the preconditioned S2. On day 7, rats were tested for their levels of freezing to the conditioned S1. For each test, the first stimulus was presented 2 min after rats were placed in the chambers. Each test consisted of eight stimulus alone presentations, with a 3 min interval between each presentation. Each S2 presentation was 30 s in duration, and each S1 presentation was 10 s. Rats remained in the chamber for an additional min after the final stimulus presentation.

### Experiments 2A and 2B

#### Context exposure

On days 1 and 2, rats received two 20 min exposures to the chambers in the manner described for the previous experiments.

#### Serial-order conditioning of S2 and S1

On day 3, rats were exposed to four S2-S1-shock sequences. For half the rats, the S2 was a sound and the S1 was a light. For the remaining rats, the S2 was a light and the S1 was a sound. Each presentation of the S2 lasted for 30 s, each presentation of the S1 lasted for 10 s, and each presentation of foot shock lasted for 0.5 s. The stimuli were presented in such a way that the offset of the 30 s S2 co-occurred with the onset of the 10 s S1, which then co-terminated with the 0.5 s foot shock. The first S2 presentation occurred 5 min after rats were placed into the chambers, the interval between each S2-S1-shock sequence was 5 min (measured from the offset of the S1 to onset of the next S2), and rats remained in the chambers for an additional 1 min after exposure to the final S2-S1-shock sequence. They were then returned to their home cages.

### Context extinction

On day 4, rats received two 20 min exposures to the chambers, one in the morning and the other 3 hr later in the afternoon. These exposures were intended to extinguish any freezing elicited by the context alone so that the level of freezing to the S2 and S1 could be assessed unconfounded by context-elicited freezing. On day 5, rats received a further 10 min extinction exposure to the context.

### Testing

On day 5, approximately 2 hr after the context extinction session, rats were tested for their levels of freezing to the S2. On day 6, rats were tested for their levels of freezing to the S1. The details of these test sessions were identical to those described for the previous experiments.

## Experiments 3A and 3B

### Context exposure

On days 1 and 2, rats received two 20 min exposures to the chambers in the manner described for Experiments 1A and 1B.

### Stage 1: Sensory preconditioning

On day 3, rats received a single session of sensory preconditioning in which they were exposed to eight paired presentations of the stimuli designated S2 and S1. For half the rats, the S2 was a sound and the S1 was a light. For the remaining rats, the S2 was a light and the S1 was a sound. The details for this session were identical to those described for the sensory preconditioning session in Experiments 1A and 1B.

### Stage 2: Serial-order conditioning of the S2 and S1

On day 4, rats received a single session in which they were exposed to four S2-S1-shock sequences. The details for this session were identical to those described for the serial-order conditioning session in Experiments 2A and 2B.

### Context extinction

On day 5, rats received two 20 min sessions of context extinction, and this was followed by an additional 10 min context extinction session on day 6. The details and purpose of these sessions were exactly as described for each of the previous experiments.

### Testing

On day 6, approximately 2 hr after the context extinction session, rats were tested for their levels of freezing to the S2. On day 7, rats were tested for their levels of freezing to the S1. The details of these test sessions were exactly as described for each of the previous experiments.

## Experiments 4A and 4B

### Context exposure

On days 1 and 2, rats received two 20 min exposures to the chambers in the manner described for Experiments 1A and 1B.

### Stage 1: Sensory preconditioning

On day 3, rats received a single session of sensory preconditioning in which they were exposed to eight paired presentations of the stimuli designated S2 and S1. For half the rats, the S2 was a sound and the S1 was a light. For the remaining rats, the S2 was a light and the S1 was a sound. The details for this session were identical to those described for the sensory preconditioning session in Experiments 1A and 1B.

### Stage 2: Trace conditioning of the S2

On day 4, rats received a single session in which they were exposed to four S2-[10 s]-shock pairings. The stimuli were presented in such a way that the offset of the 30 s S2 was followed by a 10 s trace interval, which co-terminated with a 0.5 s foot shock. The first S2 presentation occurred 5 min after

rats were placed into the chambers, the interval between each S2-[10 s]-shock pairing was 5 min, and rats remained in the chambers for an additional 1 min after exposure to the final S2-[10 s]-shock pairing. They were then returned to their home cages.

### Context extinction

On day 5, rats received two 20 min sessions of context extinction, and this was followed by an additional 10 min context extinction session on day 6. The details and purpose of these sessions were exactly as described for each of the previous experiments.

### Testing

On day 6, approximately 2 hr after the context extinction session, rats were tested for their levels of freezing to the S2. On day 7, rats were tested for their levels of freezing to the S1. The details of these test sessions were exactly as described for each of the previous experiments.

## Scoring and statistics

Conditioning and test sessions were recorded. Freezing, defined as the absence of all movements except those required for breathing (*Fanselow, 1980*), was used as a measure of conditioned fear. Freezing data were collected using a time-sampling procedure in which each rat was scored as either 'freezing' or 'not freezing' every 2 s by an observer that was unaware of the rat's group allocation. A second naïve observer also scored all of the test data. The correlation between the two observers' scores was high (Pearson's $r > 0.9$). The data were analyzed using a set of planned orthogonal contrasts (*Hays, 1963*), with the Type 1 error rate controlled at $\alpha = 0.05$. Standardized 95% confidence intervals (CIs) are reported for all significant results, and Cohen's $d$ is reported as a measure of effect size (where 0.2, 0.5, and 0.8 is a small, medium, and large effect size, respectively). The required number of rats per group was determined during the design stage of the study. It was based on our prior studies of sensory preconditioning which indicated that eight subjects per group (or per comparison in the contrast testing procedure) provides sufficient statistical power to detect effect sizes >0.5 with the recommended probability of 0.8–0.9. Please note that the details for scoring and data analysis are exactly the same as those reported in the earlier study by *Wong et al., 2025*.

## Acknowledgements

This work was supported by an Australian Government Research Training Fellowship to FSW, an Australian Research Council (ARC) Discovery Project Grant to RFW (DP2201036501), and an ARC Future Fellowship to NMH (FT190100697).

## Additional information

### Competing interests

Nathan M Holmes: Reviewing editor, eLife. The other authors declare that no competing interests exist.

### Funding

| Funder | Grant reference number | Author |
|---|---|---|
| Australian Research Council | DP2201036501 | R Fred Westbrook |
| Australian Research Council | FT190100697 | Nathan M Holmes |

The funders had no role in study design, data collection and interpretation, or the decision to submit the work for publication.

## Author contributions

Francesca S Wong, Conceptualization, Formal analysis, Investigation, Methodology, Writing - original draft, Project administration; Simon Killcross, Conceptualization, Resources, Supervision, Funding acquisition, Project administration, Writing – review and editing; R Fred Westbrook, Conceptualization, Supervision, Funding acquisition, Project administration, Writing – review and editing; Nathan M Holmes, Conceptualization, Resources, Supervision, Funding acquisition, Investigation, Project administration, Writing – review and editing

## Author ORCIDs

Francesca S Wong ⓘ https://orcid.org/0000-0002-8533-9833
Nathan M Holmes ⓘ https://orcid.org/0000-0002-0592-2026

## Ethics

All procedures were approved by the Animal Care and Ethics Committee at the University of New South Wales (iRECS 8988) and conducted in accordance with the Australian code for the care and use of animals for scientific purposes (published by the National Health and Medical Research Council of Australia). Please note that these subject details (source, housing conditions, and animal maintenance) are exactly the same as those reported in past studies by our laboratory (e.g., Kennedy et al., 2024; Wong et al., 2025).

Reviewer #1 (Public review): https://doi.org/10.7554/eLife.107943.3.sa1
Reviewer #2 (Public review): https://doi.org/10.7554/eLife.107943.3.sa2
Reviewer #3 (Public review): https://doi.org/10.7554/eLife.107943.3.sa3
Author response https://doi.org/10.7554/eLife.107943.3.sa4

---

# Additional files

## Supplementary files

MDAR checklist

## Data availability

All data generated or analysed during this study are included in the manuscript and supporting files; source data files have been provided for *Figures 1–4* via the Open Science Framework repository (https://osf.io/83pqj). These files contain the numerical data used to generate the figures.

The following previously published dataset was used:

| Author(s) | Year | Dataset title | Dataset URL | Database and Identifier |
|---|---|---|---|---|
| Wong F | 2026 | The basolateral amygdala complex and perirhinal cortex represent focal and peripheral states of information processing in rats. | https://osf.io/83pqj | Open Science Framework, 83pqj |

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
