## [Editor Report · eLife Assessment]

This **important** Research Advance builds on the authors' previous work delineating the roles of the rodent perirhinal cortex and the basolateral amygdala in first- and second-order learning. The **convincing** results show that serial exposure of non-motivationally relevant stimuli influences how those stimuli are encoded within the perirhinal cortex and basolateral amygdala when paired with a shock. This manuscript will be interesting for researchers in cognitive and behavioral neuroscience.

---

## [Referee Report · Reviewer #1 (Public review)]

Summary:

This study advances the lab's growing body of evidence exploring higher-order learning and its neural mechanisms. They recently found that NMDA receptor activity in the perirhinal cortex was necessary for integrating stimulus-stimulus associations with stimulus-shock associations (mediated learning) to produce preconditioned fear, but it was not necessary for forming stimulus-shock associations. On the other hand, basolateral amygdala NMDA receptor activity is required for forming stimulus-shock memories. Based on these facts, the authors assessed: 1. why the perirhinal cortex is necessary for mediated learning but not direct fear learning and 2. the determinants of perirhinal cortex versus basolateral amygdala necessity for forming direct versus indirect fear memories. The authors used standard sensory preconditioning and variants designed to manipulate the novelty and temporal relationship between stimuli and shock and, therefore, the attentional state under which associative information might be processed. Under experimental conditions where information would presumably be processed primarily in the periphery of attention (temporal distance between stimulus/shock or stimulus pre-exposure), perirhinal cortex NMDA receptor activation was required for learning indirect associations. On the other hand, when information would likely be processed in focal attention (novel stimulus contiguous with shock), basolateral amygdala NMDA activity was required for learning direct associations. Together, the findings indicate that the perirhinal cortex and basolateral amygdala subserve peripheral and focal attention, respectively. The authors provide support for their conclusions using careful, hypothesis-driven experimental design, rigorous methods, and integrating their findings with the relevant literature on learning theory, information processing, and neurobiology. Therefore, this work will be highly interesting to several fields.

Strengths:

(1) The experiments were carefully constructed and designed to test hypotheses that were rooted in the lab's previous work, in addition to established learning theory and information processing background literature.

(2) There are clear predictions and alternative outcomes. The provided table does an excellent job of condensing and enhancing the readability of a large amount of data.

(3) In a broad sense, attention states are a component of nearly every behavioral experiment. Therefore, identifying their engagement by dissociable brain areas and under different learning conditions is an important area of research.

(4) The authors clearly note where they replicated their own findings, report full statistical measures, effect sizes, and confidence intervals, indicating the level of scientific rigor.

(5) The findings raise questions for future experiments that will further test the authors' hypotheses; this is well discussed.

---

## [Referee Report · Reviewer #2 (Public review)]

This paper continues the authors' research on the roles of the basolateral amygdala (BLA) and the perirhinal cortex (PRh) in sensory preconditioning (SPC) and second order conditioning (SOC). In this manuscript, the authors explore how prior exposure to stimuli may influence which regions are necessary for conditioning to the second-order cue (S2). The authors perform a series of experiments which first confirm prior results shown by the author - that NMDA receptors in the PRh are necessary in SPC during conditioning of the first-order cue (S1) with shock to allow for freezing to S2 at test; and that NMDA receptors in the BLA are necessary for S1 conditioning during the S1-shock pairings. The authors then set out to test the hypothesis that the PRh encodes associations in a peripheral state of attention whereas the BLA encodes associations in a focal state of attention, similar to the A1 and A2 states in Wagner's theory of SOP. To do this, they show that BLA is necessary for conditioning to S2 when the S2 is first exposed during a serial compound procedure - S2-S1-shock. To determine whether pre-exposure of S2 will shift S2 to a peripheral focal state, the authors run a design in which S2-S1 presentations are given prior to the serial compound phase. The authors show that this restores NMDA receptor activity within the PRh as necessary for fear response to S2 at test. They then test whether the presence of S1 during the serial compound conditioning allows the PRh to support the fear responses to S2 by introducing a delay conditioning paradigm in which S1 is no longer present. The authors find that PRh is no longer required and suggest that this is due to S2 remaining in the primary focal state.

Strengths:

As with their earlier work, the authors have performed a rigorous series of experiments to better understand the roles of the BLA and PRh in the learning of first- and second-order stimuli. The experiments are well-designed and clearly presented, and the results show definitive differences in functionality between the PRh and BLA. The first experiment confirms earlier findings from the lab (and others), and the authors then build on their previous work to more deeply reveal how these regions differ in how they encode associations between stimuli. The authors have done a commendable job on pursuing these questions.

Table 1 is an excellent way to highlight the results and provide the reader with a quick look-up table of the findings.

---

## [Referee Report · Reviewer #3 (Public review)]

Summary:

This manuscript presents a series of experiments that further investigate the roles of the BLA and PRH in sensory preconditioning, with a particular focus on understanding their differential involvement in the association of S1 and S2 with shock.

Strengths:

The motivation for the study is clearly articulated, and the experimental designs are thoughtfully constructed. I especially appreciate the inclusion of Table 1, which makes the designs easy to follow. The results are clearly presented, and the statistical analyses are rigorous.

During the revision, the authors have adequately addressed my minor suggestions from the original version.

---

## [Author Response]

The following is the authors’ response to the original reviews.

**Reviewer #1 (Public review):**
Summary:This study advances the lab's growing body of evidence exploring higher-order learning and its neural mechanisms. They recently found that NMDA receptor activity in the perirhinal cortex was necessary for integrating stimulus-stimulus associations with stimulus-shock associations (mediated learning) to produce preconditioned fear, but it was not necessary for forming stimulus-shock associations. On the other hand, basolateral amygdala NMDA receptor activity is required for forming stimulus-shock memories. Based on these facts, the authors assessed: (1) why the perirhinal cortex is necessary for mediated learning but not direct fear learning, and (2) the determinants of perirhinal cortex versus basolateral amygdala necessity for forming direct versus indirect fear memories. The authors used standard sensory preconditioning and variants designed to manipulate the novelty and temporal relationship between stimuli and shock and, therefore, the attentional state under which associative information might be processed. Under experimental conditions where information would presumably be processed primarily in the periphery of attention (temporal distance between stimulus/shock or stimulus pre-exposure), perirhinal cortex NMDA receptor activation was required for learning indirect associations. On the other hand, when information would likely be processed in focal attention (novel stimulus contiguous with shock), basolateral amygdala NMDA activity was required for learning direct associations. Together, the findings indicate that the perirhinal cortex and basolateral amygdala subserve peripheral and focal attention, respectively. The authors provide support for their conclusions using careful, hypothesis-driven experimental design, rigorous methods, and integrating their findings with the relevant literature on learning theory, information processing, and neurobiology. Therefore, this work will be highly interesting to several fields.Strengths:(1) The experiments were carefully constructed and designed to test hypotheses that were rooted in the lab's previous work, in addition to established learning theory and information processing background literature.(2) There are clear predictions and alternative outcomes. The provided table does an excellent job of condensing and enhancing the readability of a large amount of data.(3) In a broad sense, attention states are a component of nearly every behavioral experiment. Therefore, identifying their engagement by dissociable brain areas and under different learning conditions is an important area of research.(4) The authors clearly note where they replicated their own findings, report full statistical measures, effect sizes, and confidence intervals, indicating the level of scientific rigor.(5) The findings raise questions for future experiments that will further test the authors' hypotheses; this is well discussed.Weaknesses:As a reader, it is difficult to interpret how first-order fear could be impaired while preconditioned fear is intact; it requires a bit of "reading between the lines".

We appreciate the Reviewer’s point and have attempted to address on lines 55-63 of the revised paper: “In a recent pair of studies, we extended these findings in two ways. First, we showed that S1 does not just form an association with shock in stage 2; it also mediates an association between S2 and the shock. Thus, S2 enters testing in stage 3 already conditioned, able to elicit fear responses (Wong et al., 2019). Second, we showed that this mediated S2-shock association requires NMDAR-activation in the PRh, as well as communication between the PRh and BLA (Wong et al., 2025). These findings raise two critical questions: (1) why is the PRh engaged for mediated conditioning of S2 but not for direct conditioning of S1; and (2) more generally, what determines whether the BLA and/or PRh is engaged for conditioning of the S1 and/or S2?”

**Reviewer #2 (Public review):**
Summary:This paper continues the authors' research on the roles of the basolateral amygdala (BLA) and the perirhinal cortex (PRh) in sensory preconditioning (SPC) and second-order conditioning (SOC). In this manuscript, the authors explore how prior exposure to stimuli may influence which regions are necessary for conditioning to the second-order cue (S2). The authors perform a series of experiments which first confirm prior results shown by the author - that NMDA receptors in the PRh are necessary in SPC during conditioning of the first-order cue (S1) with shock to allow for freezing to S2 at test; and that NMDA receptors in the BLA are necessary for S1 conditioning during the S1-shock pairings. The authors then set out to test the hypothesis that the PRh encodes associations in a peripheral state of attention, whereas the BLA encodes associations in a focal state of attention, similar to the A1 and A2 states in Wagner's theory of SOP. To do this, they show that BLA is necessary for conditioning to S2 when the S2 is first exposed during a serial compound procedure - S2-S1-shock. To determine whether pre-exposure of S2 will shift S2 to a peripheral focal state, the authors run a design in which S2-S1 presentations are given prior to the serial compound phase. The authors show that this restores NMDA receptor activity within the PRh as necessary for the fear response to S2 at test. They then test whether the presence of S1 during the serial compound conditioning allows the PRh to support the fear responses to S2 by introducing a delay conditioning paradigm in which S1 is no longer present. The authors find that PRh is no longer required and suggest that this is due to S2 remaining in the primary focal state.Strengths:As with their earlier work, the authors have performed a rigorous series of experiments to better understand the roles of the BLA and PRh in the learning of first- and second-order stimuli. The experiments are well-designed and clearly presented, and the results show definitive differences in functionality between the PRh and BLA. The first experiment confirms earlier findings from the lab (and others), and the authors then build on their previous work to more deeply reveal how these regions differ in how they encode associations between stimuli. The authors have done a commendable job of pursuing these questions.Table 1 is an excellent way to highlight the results and provide the reader with a quick look-up table of the findings.Weaknesses:The authors have attempted to resolve the question of the roles of the PRh and BLA in SPC and SOC, which the authors have explored in previous papers. Laudably, the authors have produced substantial results indicating how these two regions function in the learning of first- and second-order cues, providing an opportunity to narrow in on possible theories for their functionality. Yet the authors have framed this experiment in terms of an attentional framework and have argued that the results support this particular framework and hypothesis - that the PRh encodes peripheral and the BLA encodes focal states of learning. This certainly seems like a viable and exciting hypothesis, yet I don't see why the results have been completely framed and interpreted this way. It seems to me that there are still some alternative interpretations that are plausible and should be included in the paper.

We appreciate the Reviewer’s point and have attempted to address it on lines 566-594 of the Discussion: “An additional point to consider in relation to Experiments 3A, 3B, 4A and 4B is the level of surprise that rats experienced following presentations of the familiar S2 in stage 2. Specifically, in Experiments 3A and 3B, S2 was followed by the expected S1 (low surprise) and its conditioning required activation of NMDA receptors in the PRh and not the BLA. By contrast, in Experiments 4A and 4B, S2 was followed by omission of the expected S1 (high surprise) and its conditioning required activation of NMDA receptors in the BLA and not the PRh. This raises the possibility that surprise, or prediction error, also influences the way that S2 is processed in focal and peripheral states of attention. When prediction error is low, S2 is processed in the peripheral state of attention: hence, learning under these circumstances requires NMDA receptor activation in the PRh and not the BLA. By contrast, when prediction error is high, S2 is preserved in the focal state of attention: hence, learning under these circumstances requires NMDA receptor activation in the BLA and not the PRh. The impact of prediction error on the processing of S2 could be assessed using two types of designs. In the first design, rats are pre-exposed to S2-S1 pairings in stage 1 and this is followed by S2-S3-shock pairings in stage 2. The important feature of this design is that, in stage 2, the S2 is followed by surprise in omission of S1 and presentation of S3. Thus, if a large prediction error maintains processing of the familiar S2 in the BLA, we might expect that its conditioning in this design would require NMDA receptor activation in the BLA (in contrast to the results of Experiment 3B) and no longer require NMDA receptor activation in the PRh (in contrast to the results of Experiment 3A). In the second design, rats are pre-exposed to S2 alone in stage 1 and this is followed by S2-[trace]-shock pairings in stage 2. The important feature of this design is that, in stage 2, the S2 is not followed by the surprising omission of any stimulus. Thus, if a small prediction error shifts processing of the familiar S2 to the PRh, we might expect that its conditioning in this design would no longer require NMDA receptor activation in the BLA (in contrast to the results of Experiment 4B) but, instead, require NMDA receptor activation in the PRh (in contrast to the results of Experiment 4A). Future studies will use both designs to determine whether prediction error influences the processing of S2 in the focus versus periphery of attention and, thereby, whether learning about this stimulus requires NMDA receptor activation in the BLA or PRh.”

**Reviewer #3 (Public review):**
Summary:This manuscript presents a series of experiments that further investigate the roles of the BLA and PRH in sensory preconditioning, with a particular focus on understanding their differential involvement in the association of S1 and S2 with shock.Strengths:The motivation for the study is clearly articulated, and the experimental designs are thoughtfully constructed. I especially appreciate the inclusion of Table 1, which makes the designs easy to follow. The results are clearly presented, and the statistical analyses are rigorous. My comments below mainly concern areas where the writing could be improved to help readers more easily grasp the logic behind the experiments.Weaknesses:(1) Lines 56-58: The two previous findings should be more clearly summarized. Specifically, it's unclear whether the "mediated S2-shock" association occurred during Stage 2 or Stage 3. I assume the authors mean Stage 2, but Stage 2 alone would not yet involve "fear of S2," making this expression a bit confusing.

We apologise for the confusion and have revised the summary of our previous findings on lines 55-63. The revised text now states: “In a recent pair of studies, we extended these findings in two ways. First, we showed that S1 does not just form an association with shock in stage 2; it also mediates an association between S2 and the shock. Thus, S2 enters testing in stage 3 already conditioned, able to elicit fear responses (Wong et al., 2019). Second, we showed that this mediated S2-shock association requires NMDAR-activation in the PRh, as well as communication between the PRh and BLA (Wong et al., 2025). These findings raise two critical questions: (1) why is the PRh engaged for mediated conditioning of S2 but not for direct conditioning of S1; and (2) more generally, what determines whether the BLA and/or PRh is engaged for conditioning of the S1 and/or S2?”

(2) Line 61: The phrase "Pavlovian fear conditioning" is ambiguous in this context. I assume it refers to S1-shock or S2-shock conditioning. If so, it would be clearer to state this explicitly.

Apologies for the ambiguity - we have omitted the term “Pavlovian” which may have been the source of confusion: The revised text on lines 60-63 now states: “These findings raise two critical questions: (1) why is the PRh engaged for mediated conditioning of S2 but not for direct conditioning of S1; and (2) more generally, what determines whether the BLA and/or PRh is engaged for conditioning of the S1 and/or S2?”

(3) Regarding the distinction between having or not having Stage 1 S2-S1 pairings, is "novel vs. familiar" the most accurate way to frame this? This terminology could be misleading, especially since one might wonder why S2 couldn't just be presented alone on Stage 1 if novelty is the critical factor. Would "outcome relevance" or "predictability" be more appropriate descriptors? If the authors choose to retain the "novel vs. familiar" framing, I suggest providing a clear explanation of this rationale before introducing the predictions around Line 118.

We have incorporated the suggestion regarding “predictability” while also retaining “novelty” as follows.

L76-85: “For example, different types of arrangements may influence the substrates of conditioning to S2 by influencing its novelty and/or its predictive value at the time of the shock, on the supposition that familiar stimuli are processed in the periphery of attention and, thereby, the PRh (Bogacz & Brown, 2003; Brown & Banks, 2015; Brown & Bashir, 2002; Martin et al., 2013; McClelland et al., 2014; Morillas et al., 2017; Murray & Wise, 2012; Robinson et al., 2010; Suzuki & Naya, 2014; Voss et al., 2009; Yang et al., 2023) whereas novel stimuli are processed in the focus of attention and, thereby, the amygdala (Holmes et al., 2018; Qureshi et al., 2023; Roozendaal et al., 2006; Rutishauser et al., 2006; Schomaker & Meeter, 2015; Wright et al., 2003).”

L116-120: “Subsequent experiments then used variations of this protocol to examine whether the engagement of NMDAR in the PRh or BLA for Pavlovian fear conditioning is influenced by the novelty/predictive value of the stimuli at the time of the shock (second implication of theory) as well as their distance or separation from the shock (third implication of theory; Table 1).”

(4) Line 121: This statement should refer to S1, not S2.(5) Line 124: This one should refer to S2, not S1.

We have checked the text on these lines for errors and confirmed that the statements are correct. The lines encompassing this text (L121-130) are reproduced here for convenience:

(1) When rats are exposed to novel S2-S1-shock sequences, conditioning of S2 and S1 will be disrupted by a DAP5 infusion into the BLA but not into the PRh (Experiments 2A and 2B);

(2) When rats are exposed to S2-S1 pairings and then to S2-S1-shock sequences, conditioning of S2 will be disrupted by a DAP5 infusion into the PRh but not the BLA whereas conditioning of S1 will be disrupted by a DAP5 infusion into the BLA not the PRh (Experiments 3A and 3B);

(3) When rats are exposed to S2-S1 pairings and then to S2 (trace)-shock pairings, conditioning of S2 will be disrupted by a DAP5 into the BLA not the PRh (Experiments 4A and 4B).

(6) Additionally, the rationale for Experiment 4 is not introduced before the Results section. While it is understandable that Experiment 4 functions as a follow-up to Experiment 3, it would be helpful to briefly explain the reasoning behind its inclusion.

Experiment 4 follows from the results obtained in Experiment 3; and, as noted, the reasoning for its inclusion is provided locally in its introduction. We attempted to flag this experiment earlier in the general introduction to the paper; but this came at the cost of clarity to the overall story. As such, our revised paper retains the local introduction to this experiment. It is reproduced here for convenience:

“In Experiments 3A and 3B, conditioning of the pre-exposed S1 required NMDAR-activation in the BLA and not the PRh; whereas conditioning of the pre-exposed S2 required NMDAR-activation in the PRh and not the BLA. We attributed these findings to the fact that the pre-exposed S2 was separated from the shock by S1 during conditioning of the S2-S1-shock sequences in stage 2: hence, at the time of the shock, S2 was no longer processed in the focal state of attention supported by the BLA; instead, it was processed in the peripheral state of attention supported by the PRh.

“Experiments 4A and 4B employed a modification of the protocol used in Experiments 3A and 3B to examine whether a pre-exposed S1 influences the processing of a pre-exposed S2 across conditioning with S2-S1-shock sequences. The design of these experiments is shown in Figure 4A. Briefly, in each experiment, two groups of rats were exposed to a session of S2-S1 pairings in stage 1 and, 24 hours later, a session of S2-[trace]-shock pairings in stage 2, where the duration of the trace interval was equivalent to that of S1 in the preceding experiments. Immediately prior to the trace conditioning session in stage 2, one group in each experiment received an infusion of DAP5 or vehicle only into either the PRh (Experiment 4A) or BLA (Experiment 4B). Finally, all rats were tested with presentations of the S2 alone in stage 3. If the substrates of conditioning to S2 are determined only by the amount of time between presentations of this stimulus and foot shock in stage 2, the results obtained in Experiments 4A and 4B should be the same as those obtained in Experiments 3A and 3B: acquisition of freezing to S2 will require activation of NMDARs in the PRh and not the BLA. If, however, the presence of S1 in the preceding experiments (Experiments 3A and 3B) accelerated the rate at which processing of S2 transitioned from the focus of attention to its periphery, the results obtained in Experiments 4A and 4B will differ from those obtained in Experiments 3A and 3B. That is, in contrast to the preceding experiments where acquisition of freezing to S2 required NMDAR-activation in the PRh and not the BLA, here acquisition of freezing to S2 should require NMDAR-activation in the BLA but not the PRh.”

**Reviewer #1 (Recommendations for the authors):**
I greatly enjoyed reading and reviewing this manuscript, and so I only have boilerplate recommendations.(1) I might add a couple of sentences discussing how/why preconditioned fear could be intact while first-order fear is impaired. Of course, if I am interpreting the provided interpretation correctly, the reason is that peripheral processing is still intact even when BLA NMDA receptors are blocked, and so mediated conditioning still occurs. Does this mean that mediated conditioning does not require learning the first-order relationship, and that they occur in parallel? Perhaps I just missed this, but I cannot help but wonder whether/how the psychological processes at play might change when first-order learning is impaired, so this would be greatly appreciated.

As noted above, we have revised the general introduction (around lines 55-59) to clarify that the direct S1-shock and mediated S2-shock associations form in parallel. Hence, manipulations that disrupt first-order fear to the S1 (such as a BLA infusion of the NMDA receptor antagonist, DAP5) do not automatically disrupt the expression of sensory preconditioned fear to the S2.

(2) Adding to the above - does the SOP or another theory predict serial vs parallel information flow from focal state to peripheral, or perhaps it is both to some extent?

SOP predicts both serial and parallel processing of information in its focal and peripheral states. That is, some proportion of the elements that comprise a stimulus may decay from the focal state of attention to the periphery (serial processing); hence, at any given moment, the elements that comprise a stimulus can be represented in both focal and peripheral states (parallel processing).

Given the nature of the designs and tools used in the present study (between-subject assessment of a DAP5 effect in the BLA or PRh), we selected parameters that would maximize the processing of the S2 and S1 stimuli in one or the other state of activation; hence the results of the present study. We are currently examining the joint processing of stimulus elements across focal and peripheral states using simultaneous recordings of activity in the BLA and PRh. These recordings are collected from rats trained in the different stages of a within-subject sensory preconditioning protocol. The present study created the basis for this work, which will be published separately in due course.

(3) The organization of PRh vs BLA is nice and consistent across each figure, but I would suggest adding any kind of additional demarcation beyond the colors and text, maybe just more space between AB / CD. The figure text indicating PRh/BLA is a bit small.

Thank you for the suggestion – we have added more space between the top and bottom panels of the figure.

(4) Line 496 typo ..."in the BLA but not the BLA".

Apologies for the type - this has been corrected.

**Reviewer #2 (Recommendations for the authors):**
I found the experiments to be extremely well-designed and the results convincing and exciting. The hypothesis of the focal and peripheral states of attention being encoded by BLA and PRh respectively, is enticing, yet as indicated in the public review, this does not seem to be the only possible interpretation. This is my only serious comment for the authors.(1) I think it would be worth reframing the article slightly to give credence to alternative hypotheses. Not to say that the authors' intriguing hypothesis shouldn't be an integral part of the introduction, but no alternatives are mentioned. In experiment 2, could the fact that S2 is already being a predictor of S1, not block new learning to S2? In the framework of stimulus-stimulus associations, there would be no surprise in the serial-compound stage of conditioning at the onset of S1. This may prevent direct learning of the S2-shock association within the BLA. This type of association may as well (S2 predicts S1, but it's omitted), which could support learning by S2. fall under the peripheral/focal theory, but I don't think it's necessary to frame this possibility in terms of a peripheral/focal theory. To build on this alternative interpretation, the absence of S1 in experiment 4 may induce a prediction error. The peripheral and focal states appear to correspond to A2 and A1 in SOP extremely well, and I think it would potentially add interest and support. If the authors do intend to make the paper a strong argument for their hypothesis, perhaps a few additional experiments may be introduced. If the novelty of S2 is critical for S2 not to be processed in a focal state during the serial compound stage, could pre-exposure of S2 alone allow for dependence of S2-shock on the PRh? Assuming this is what the authors would predict, this might disentangle the S-S theory mentioned above from the peripheral/focal theory. Or perhaps run an experiment S2-X in stage 1 and S2-S1-shock in stage 2? This said, I think the experiments are more than sufficient for an exciting paper as is, and I don't think running additional experiments is necessary. I would only argue for this if the authors make a hard claim about the peripheral/focal theory, as is the case for the way the paper is currently written.

We appreciate the reviewer’s excellent point and suggestions. We have included an additional paragraph in the Discussion on page 24 (lines 566-594). “An additional point to consider in relation to Experiments 3A, 3B, 4A and 4B is the level of surprise that rats experienced following presentations of the familiar S2 in stage 2. Specifically, in Experiments 3A and 3B, S2 was followed by the expected S1 (low surprise) and its conditioning required activation of NMDA receptors in the PRh and not the BLA. By contrast, in Experiments 4A and 4B, S2 was followed by omission of the expected S1 (high surprise) and its conditioning required activation of NMDA receptors in the BLA and not the PRh. This raises the possibility that surprise, or prediction error, also influences the way that S2 is processed in focal and peripheral states of attention. When prediction error is low, S2 is processed in the peripheral state of attention: hence, learning under these circumstances requires NMDA receptor activation in the PRh and not the BLA. By contrast, when prediction error is high, S2 is preserved in the focal state of attention: hence, learning under these circumstances requires NMDA receptor activation in the BLA and not the PRh. The impact of prediction error on the processing of S2 could be assessed using two types of designs. In the first design, rats are pre-exposed to S2-S1 pairings in stage 1 and this is followed by S2-S3-shock pairings in stage 2. The important feature of this design is that, in stage 2, the S2 is followed by surprise in omission of S1 and presentation of S3. Thus, if a large prediction error maintains processing of the familiar S2 in the BLA, we might expect that its conditioning in this design would require NMDA receptor activation in the BLA (in contrast to the results of Experiment 3B) and no longer require NMDA receptor activation in the PRh (in contrast to the results of Experiment 3A). In the second design, rats are pre-exposed to S2 alone in stage 1 and this is followed by S2-[trace]-shock pairings in stage 2. The important feature of this design is that, in stage 2, the S2 is not followed by the surprising omission of any stimulus. Thus, if a small prediction error shifts processing of the familiar S2 to the PRh, we might expect that its conditioning in this design would no longer require NMDA receptor activation in the BLA (in contrast to the results of Experiment 4B) but, instead, require NMDA receptor activation in the PRh (in contrast to the results of Experiment 4A). Future studies will use both designs to determine whether prediction error influences the processing of S2 in the focus versus periphery of attention and, thereby, whether learning about this stimulus requires NMDA receptor activation in the BLA or PRh.”

(3) I was surprised the authors didn't frame their hypothesis more in terms of Wagner's SOP model. It was minimally mentioned in the introduction or the authors' theory if it were included more in the introduction. I was wondering whether the authors may have avoided this framing to avoid an expectation for modeling SOP in their design. If this were the case, I think the paper stands on its own without modeling, and at least for myself, a comparison to SOP would not require modeling of SOP. If this was the authors' concern for avoiding it, I would suggest to the authors that they need not be concerned about it.

We appreciate the endorsement of Wagner’s SOP theory as a nice way of framing our results. We are currently working on a paper in which we use simulations to show how Wagner’s theory can accommodate the present findings as well as others in the literature on sensory preconditioning. For this reason, we have not changed the current paper in relation to this point.